# Evolution of the Materials and Methods Used for Subsurface Drainage of Agricultural Lands from Antiquity to the Present

**Stavros I. Yannopoulos [1,*], Mark E. Grismer [2], Khaled M. Bali [3] and Andreas N. Angelakis [4,5]**

[1] Faculty of Engineering, School of Rural and Surveying Engineering, Aristotle University of Thessaloniki, 54124 Thessaloniki, Greece

[2] Departments of LAWR and Biological & Agricultural Engineering, UC Davis, Davis, CA 95616, USA; megrismer@ucdavis.edu

[3] Division of Agriculture and Natural Resources—Kearney Agricultural Research and Extension Center, University of California, Parlier, CA 93648, USA; kmbali@ucanr.edu

[4] HAO-Demeter, Agricultural Research Institution of Crete, 71300 Iraklion, Greece; angelak@edeya.gr

[5] Union of Hellenic Water Supply and Sewerage Operators, 41222 Larissa, Greece

**\*** Correspondence: giann@vergina.eng.auth.gr; Tel.: +30-2310-996114

**Abstract:** Agricultural drainage plays an important role worldwide in food production and conservation of soil resources, while safeguarding investments in agricultural production and irrigation projects. It can improve crop yields and land productivity, especially on poorly drained soils and in cases of prolonged waterlogging. Both the subsurface drainage materials and the installation techniques used have a long history dating to prehistoric times. Over time, new subsurface drainage materials, installation techniques and modernized equipment were being developed continuously to take advantage of technological advances provided through research and development, while the planning and organization of the implementation process were improved. Today's new materials and improved installation methods can offer solutions to problems still unsolved, while sometimes creating new ones. This paper considers the evolution of basic subsurface drainage materials and their installation techniques as they developed and adapted over time as well as possible future trends in drainage system design and application.

**Keywords:** subsurface drainage; drain envelope; drainage pipe; installation equipment; agricultural land; drainage machinery

## 1. Introduction

The importance of adequate drainage of agricultural lands has long been recognized, given the fact that drainage is probably as old as agriculture itself, dating from antiquity. In antiquity, serious drainage problems occurred in irrigated areas in several regions of the world, just as today. This is probably the reason for the decline and disappearance of some ancient irrigation-based civilizations (e.g., Mesopotamians) as they failed to confront the hazards due to poor land drainage [1]. Nowadays, in several regions of the world, some agricultural lands (e.g., coastal plains, river valleys, and inland plains) have limited natural drainage capacity due to poorly drained root zone soils and or shallow water tables. To provide a root environment suitable for maximum crop production in these conditions, artificial drainage is required to control root zone waterlogging and salt accumulation, or to prevent surface flooding during periods of high rainfall [2]. More specifically, land drainage is part of agricultural water management to enhance crop growth and maintain soil

productivity. In particular, the direct goals of a land drainage system are: (a) removal of the excess of surface and subsurface water; (b) maintenance of groundwater levels for adequate root zone aeration and trafficability; (c) removal of excess soluble salts from the root zone associated with the applied irrigation water or upward flow shallow groundwater.

The problem of land drainage associated with agriculture was closely analyzed and studied by the Roman writers Cato (Marco Porcius Cato, 234 BC–149 BC) in his "De Agricultura" (On Farming), Varo (Marcus Terentius Varo, 116 BC–28 BC) in his "Rerum Rusticarum Libri Tres" (Three Books on Agriculture), Polio (Marcus Vitruvius Polio, 80 BC–26 AD) in his "De Architectura Libri Decern" (Architecture in Ten Books), Collumela (Lucius Junius Moderatus Columella, 4 AD–ca. 70 AD) in his "De Re Rustica" (On Agriculture), Plinius (Gaios Plinius Secundus, ca. 23 AD–79 AD) in his "Historia Naturalis" (Natural History), Frontinus (Sextus Julius Frontinus, ca. 35 AD–103 AD), Palladius (Palladius Rutillius Taurus Aemilianus, late in the 4th century or first half of the 5th century AD) on his "Opus Agriculturae" (The Work of Farming).

All these writers mentioned drainage, and some of them gave minute directions for forming drains with stones, branches of trees, and straw. For example, Cato directed drains to be made through shaped ditches dug three feet broad at the top, four feet deep, and one foot and a quarter wide at the bottom, to be filled with stones or with willow rods placed crosswise, or twigs tied together. Columella stated covered drains are to be made sloping at the sides, and the bottom to be made narrow. In addition, he recommended a rope of twigs to be firmly pressed into the drain and covered with leaves or pine branches before filling in. Plinius mentioned the ropes may be of straw, and that flint or gravel may moreover be used for the waterway, the excavation being filled to within eighteen inches of the top.

Finally, Palladius described the agricultural land drainage as follows [3]: "*When the lands are wet, they will be dried up by digging trenches everywhere. Everyone knows how to make open trenches, but here is the way to make hidden trenches: One digs out across the field ditches of three feet in depth, which are to be half filled with small stones or gravel; after which they are filled up with the earth from the digging und leveled. However, the ends of those causeways must lead in declivity unto an open ditch whither the water will run without carrying away the earth of the field. Should there be no stones; one will lay at the bottom of the ditch fascines, straw, or briars of any kind whatever.*" As Miles [4] stated, "*these old Romans were the sole authorities on draining and their methods were practiced, without any improvement, for more than a thousand years*".

Recently, Valipour et al. [5] presented a review that traces the evolution of the main founding of agricultural drainage technologies through the centuries until the present. This historical review reveals valuable insights into ancient hydraulic technologies and the management of irrigation and drainage over the years.

There are two major types of drainage systems for controlling water table levels and for removing excess water from a field, namely open ditches of various sizes and shapes (surface drainage) and subsurface piping (subsurface drainage). The choice depends on the type of drainage problem encountered and the soil's physical properties. Consequently, the solution may be a surface drainage system, a subsurface drainage system, or as is often the case, a combination of the two.

Often, the general term "*agricultural drainage*" is used for both surface and subsurface drainage. In the first case (surface drainage), open drains are nearly always applied, which entails the collection of surface runoff before saturation of the root zone. In the second case (subsurface drainage), the removal of excess water from the root zone is achieved by lowering the groundwater level and it is accomplished by means of open ditches, by tube drains or by a combination of both. At first, this technique was called "*tile drainage*" because cylindrical tiles were laid end to end in a trench. Open drains, to a certain extent, support the controlling of the water table. However, they present serious disadvantages, for example, loss of productive agricultural land, hindrances for farming operations and overland traffic, and heavy maintenance requirements due to weed growth and sediment buildup or instability of banks and consequently, high cost of maintenance. Meanwhile, per unit area, subsurface drainage systems are generally more expensive than surface drainage. Nevertheless,

nowadays, open drains are mostly restricted to the disposal and transport of drainage water and sometimes its storage.

Subsurface drainage was developed mainly in the temperate climatic zones (Europe, North America, Russia) to control water table level, and today, it is used in semi-arid and arid zones as an integral part of the irrigated agriculture system [6]. The proper and lasting performance of subsurface drainage systems requires selection of appropriate materials (i.e., pipes and envelopes), adequate installation and their maintenance [7]. Past technologies and practices can be a useful guide towards future designs. During the 1950s and thereafter, there were rapid developments both in installation techniques and materials. These developments have been interdependent in that new materials prompted the development of new installation techniques, and vice versa. The high speed mechanical installation of subsurface drains by modern specialized machines, the development of new drainage materials, the increase in installation capacities, the reduction in installation costs and the graduated evolution from a practice based on local experience into an art with more general applicability contributed to the development of the agricultural land drainage all over the world [5].

For a successful subsurface drainage design, the knowledge of drain materials (drainage conveyance conduit) and installation techniques of the drains (including selection of envelope materials with respect to soil bedding) is a crucial element for achieving sustainable development of both irrigated and rainfed agriculture. For centuries, engineers and inventors tried to develop rapid and low cost techniques for subsurface drainage. Though many ideas were considered, only a few found widespread applications. The selection of the appropriate drainage materials (i.e., pipes and envelopes) depends mainly on their availability, durability, and cost [8]. Over time, new drainage materials, installation techniques and modernized equipment developed continuously to take advantage of technological advances provided through research and development, while the planning and organization of the implementation process were improved.

A great variety of methods and materials have been used in the drainage of agricultural lands during different times, many of them were effective and all of them aiding to demonstrate drainage usefulness in agriculture. As we go forward to the future, it is important to consider what is, in the present day, the situation of the materials used for drainage of agricultural lands and how it has evolved, since past technologies and practices can be a useful guide towards future designs. However, an extended presentation of the history and annotation of the evolution of subsurface drainage materials and installation techniques is quite broad; consequently, our focus is restricted to the main times in which these were developed for the speedy extension of agricultural drainage worldwide. We note that the literature cited and taken into account herein is not considered to be exhaustive; however, we try to recognize key scientific articles whenever possible.

The purpose of this article is to provide a succinct review and discussion of the evolution of key achievements in the development of drainage materials and installation techniques from antiquity to present times, with specific reference to subsurface drainage, considering the major materials used and lessons learned from the past. As such, our hypothesis is that a careful review and consideration of the past developments in drainage systems underscore their importance to agricultural production in many parts of the world, and moreover, that failure to adequately consider possible drainage issues when developing agricultural production may lead to its eventual demise and damage to the local ecosystem.

This review study is organized as follows: Section 1 is an introductory one; Sections 2–4 present the evolution of drainpipe materials, envelope technology, and installation technology through the centuries; Section 5 refers to the drainage system design/installation effects on drain-water quality; Section 6 addresses the alternative drainage technique to control waterlogging and salinity and in particular, biodrainage; finally, Section 7 deals with the general observations about how drainage has evolved and its importance to agricultural production.

## 2. Evolution of Drainpipe Materials

A surface drainage system is composed of field drains that collect excess water from the land surface and root zone, conduct it to lateral ditches, which then flow to a main ditch, and finally, the

drainage outlet, usually a natural waterway [9]. Similarly, a subsurface drainage system may use buried perforated pipes or tiles that collect drainage water and move it through the underground pipe system to the main drains that provide an outlet for the water captured from the root zone [10]. In general, subsurface drainage of agricultural lands likely began with the excavation of relatively shallow open ditches that accepted surface runoff and lateral discharge of shallow groundwater. In this way, these drains operated, at the same time, like a system of surface and subsurface drainage. Over the centuries, drainage pipes have been made from wood boards or box drains, bricks, ceramic tiles, clay tiles, concrete tiles, bituminized fiber perforated pipes, rigid plastic perforated pipes, corrugated wall metal conduits, perforated smooth plastic pipes or corrugated plastic pipes. In fact, historically, a "*subsurface drainage system*" was referred to as a "*tile drainage system*" due to the widespread use of buried clay/ceramic tile pipes for drainage systems. Nowadays, corrugated plastic pipes are frequently used, although clay and concrete pipes installed in the 1940s and 1950s are still used in irrigated agriculture. In any case, their application is determined by local economic factors. Undoubtedly, in the last several decades, there have been several advances in drainage installation equipment, materials, techniques, and technology, which caused changes in installation quality in many parts of the world. Currently, corrugated plastic pipes are commonly used in irrigated agriculture in most regions of the world.

## 2.1. Drainage Pipes over Antiquity

Drainage systems have a long history of application, when agricultural drainage appeared for the first time in the Mesopotamia, approximately 9000 years ago; pipes were nonexistent [11]. Due to the lack of the drainage pipes, the early subsurface drains probably were trenches partially filled with highly permeable materials, such as gravel and stones, or permeable, voluminous substances such as bundles of small trees and shrubs tied together at the bottom of the trench [8]. Perhaps the oldest known drainpipes were discovered in the Lower Indus river valley, which were fabricated about 2000 BC [12], or are approximately 4000 years old. Additionally, underground bamboo sticks with holes were used as drains in ancient times in China [13]. About 400 BC, the Egyptians and Greeks drained land using a system of surface ditches to drain individual land parcels [14].

## 2.2. Drainage Pipes from Antiquity Until About 1700

The Romans used open drains to remove ponded surface water and closed drains for removing surplus water from the soil itself. In addition, it seems they have applied clay-ware tiles for drainage purposes [15]. Clay-ware tiles dating from the 1st century AD are shown in Figure 1.

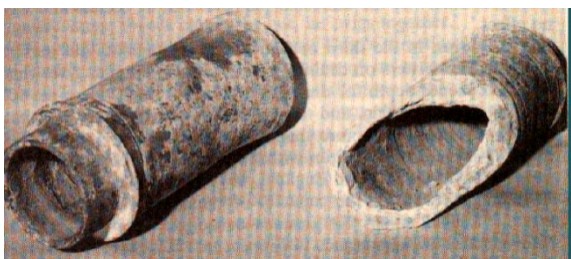

**Figure 1.** Roman clay ware tiles [16].

As Schwab and Fouss [17] pointed out, the first subsurface drain materials were probably bundles of brush tied together at the bottom of a trench or gravel and stones.

In Europe, clay tiles have been used since the beginning of Christian Era [18]. By placing a modified form of medieval clay roofing tiles in a trench, French farmers are considered the first to use clay roofing tiles for farm drainage purposes. This type of tile drainage was used at least into the 14th or 15th centuries [14]. In addition, subsurface drainage conduits made of sticks, brush and stones were generally used in Europe until at least the end of the 17th century [14]. Davidson [19] stated that land drainage by means of tile was introduced in Europe as early as 1620, but it did not come into general use until about 1850. The tile drainage pipes discovered in 1620 were located in the garden

of a convent in Maubeuge, France. Each tile was about 25.40 cm (10 in) long and 10.14 cm (4 in) in diameter. The tiles had been laid in the soil in such a manner as to form a drain system at a depth of 121.92 cm (4 feet). Each tile at one end had formed into a funnel-shape, and the other tapered into a cone (Figure 2). They were made by hand of an argilo-siliceous clay, which was very hard, then glazed in burning, and turned on a lathe [3]. It is not exactly known when these tiles were placed there [20]. Tile making machines were introduced in about 1848, and from this time on, tile drainage increased rapidly.

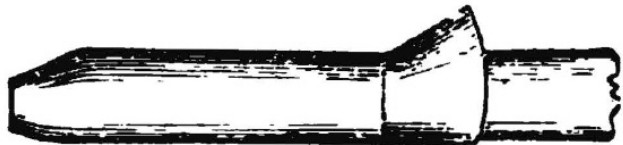

**Figure 2.** Pipe drainage discovered in Maubeuge, France in 1620 [3].

During the 17th century, installation of the subsurface drainage system in the United Kingdom included trenches filled with bushes or stones that were further developed on a larger scale across the Netherlands [21].

In the Midwestern United States (Midwest), subsurface drainage lines began to be installed in the late 1800s (earlier in New England). Most of these tile drains were pipes or sections of clay, concrete, or wood that were installed between 1870 and 1920, and again, between 1945 and 1960 [22,23].

*2.3. Drainage Pipes from 1700 Until 1940s*

Whilst the clay working industry is an ancient industry dating to unrecorded history, the first clay drainpipes were used on the estate of Sir James Graham in Northumberland in England in 1810, and they appear to have been the standard tile for about thirty years [20,24]. The invention of clay pipes was marked an important epoch in the history of drainage [23]. The earliest form of tiles introduced for drainage purposes was the horseshoe shaped tile, so called due from its shape. Sometimes, these tiles were used without any sole at the bottom of the drain (Figure 3). In this way, the water can run freely on the ground. The tiles formed a simple arch or tunnel when placed open side down and end-to-end along the bottom of a trench, and then, they were covered with excavated soil. However, this form of drain tile was short-lived because the drainage systems failed in time.

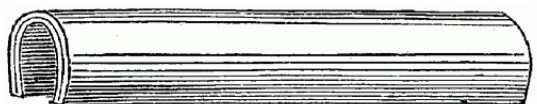

**Figure 3.** Horseshoe tile [25].

Later, these horseshoe tiles were placed on a flat piece of burned clay or pallet (sole) running the length of the tile and a width one inch wider than the tile. The pallet was used to stabilize the trench bottom soils and to create a closed conduit for the water to flow through (Figure 4). These tiles were handmade, and consequently, very costly [14]. Weaver [18] reported on the use of horseshoe tiles laid in 1760 on the Granbury estate in Suffolk, England.

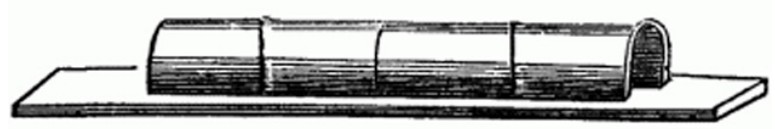

**Figure 4.** Horseshoe shaped tiles and soles properly placed [26].

Clay pipes were made in a variety of shapes, as shown in Figure 5. Moreover, many shapes of tiles and pipes can be found in the catalogue of the Museum of English Rural Life, donated by the Ministry of Agriculture Fisheries and Food Land Drainage and Water Supplies Division [27].

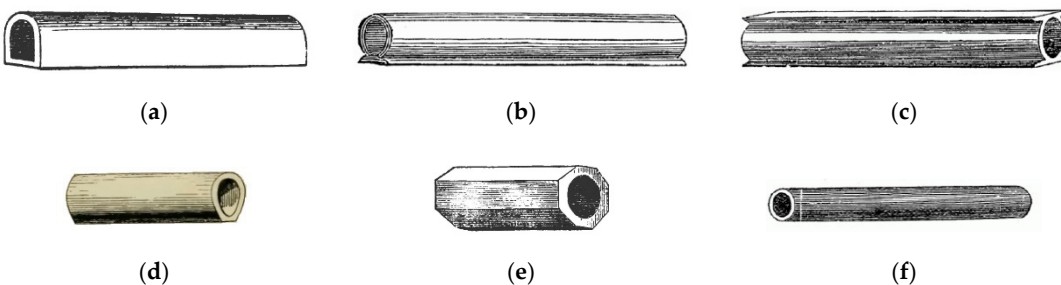

(**a**)             (**b**)             (**c**)

(**d**)             (**e**)             (**f**)

**Figure 5.** Clay pipes of variety of shapes: (**a**) Horseshoe tile and sole in one [3]; (**b**) Sole tile [25]; (**c**) Double sole tile [25]; (**d**) Tile with a flat base [28]; (**e**) Octagonal pipe tile [3] and (**f**) Cylindrical drainage tile [25]. (Processing has been made by Stavros I. Yannopoulos).

Klippart [3] stated horseshoe tiles (Figure 4) were the most widely used for a long time because of the ease in laying them. Chamberlain [29] pointed out, "*the cylindrical tiles without collars or joints, have virtually superseded all other shapes of tiles, such as the horseshoe tiles, the sole tiles, the socket or collar tiles, the oval tiles, etc.,*". He considered the cylindrical tiles "*as cheaper, stronger, better in all respects than any other; and for ordinary drainage there is no need of sockets, joints, or collars*". In addition, he recommended the purchase of "*the fully cylindrical or the octagonal outside and cylindrical inside*" that is made nearest and costs the least, if equally good otherwise. In 1903, Elliott [30] noted, "*the best shaped tile is that in which the cross section is a circle, because can be laid more easily and give greater capacity for the material used*". Moreover, he recommended clay as suitable material for the construction of drains; that is, "*tiles of circular cross section, of good clay, well burned, smooth and true in shape*". Eventually, the full cylindrical shape (cylindrical inside-outside) prevailed and was applied as a subterranean drainage conduit.

Cylindrical drainage pipes were first manufactured by John Reade, a gardener in the village of Horsemonden in the county of Kent in England in 1810. His handmade tiles were a great improvement over the old brush and stone drains and proved more popular than horseshoe drains [14]. However, no special attention was given to these pipes until they were exhibited by John Reade in the Royal Agricultural Society's Show in Derby, England, in 1843 [26,31]. As Mitchell [32] stated, while the cylindrical pipes were being used locally some twenty years earlier (about since 1823), it is thought they were first introduced about the year 1843.

The revolution in drainage construction dates back to 1845, when Thomas Scragg invented an extruding machine that produced round clay pipes quickly, reducing significantly their cost of production [14,33–36]. This invention constituted an important factor in spreading the use of the drainage worldwide. From England, the mechanical production of clay drainpipes spread over Europe and to the USA in the mid-19th century [37]. For the next century, ceramic (tile) pipes became the basic means of drainage in several countries, such as Jordan, Spain, Pakistan, and a number of other countries.

In 1830, for the first time, Portland cement was used to make drain tiles by hand [1]. The manufacturing of concrete drainpipes started in the USA in 1862, when David Ogden developed a suitable machine for making drainpipes from cement and sand [19]. This machine could make pipe with an inside diameter of 5.715 cm (2.25 in) to 60.96 cm (24 in). Obviously, the use of concrete drain tiles dominated in areas where good clay was not available, while in the other areas, economic factors determined the choice of one of these materials.

For many years, both clay and concrete pipes were the main drainage materials for the practice of subsurface drainage systems, and they were used extensively in irrigated areas. However, as Fouss [38] pointed out, "*these products today are much improved over the tile used in the early 1900s*". The choice between the two tubes is largely dependent on the delivery price at the place of installation [2].

Some disadvantages of these drainpipes are: (a) the production of clay and concrete drainpipes needed considerable labor, and (b) the protection of drainpipes against sedimentation with envelope materials is difficult to apply. However, the durability of clay pipes was very important, as they can be used in almost all circumstances. Use of concrete drainpipes in acid and high sulfate soils requires special highly resistant cement and high density concrete should be used in order to withstand sulfate degradation of the pipes.

*2.4. Drainage Pipes from 1940s Until Todays*

In the 1940s, an interesting breakthrough in pipe drainage technology took place, when rigid plastic and bituminous fiber pipes were introduced [13]. This was followed by the 1960s, when smooth plastic drainpipes were introduced that never found widespread use due to the introduction of corrugated plastic drainpipes in 1963, which overcame the disadvantages of the smooth plastic pipes. Gradually, corrugated plastic drainpipes started to replace the clay and concrete drainpipes [8]. Since the 1980s, corrugated high density polyethylene (HDPE) and corrugated polyvinyl chloride (PVC) pipes are the preferred standards for drainage pipes [13,39].

Nowadays, the corrugated plastic drains are made of PVC, high density polyethylene (HDPE) and polypropylene (PP). Since the 1980s, corrugated high density polyethylene (HDPE) and corrugated polyvinyl chloride (PVC) pipes are the preferred standards for drainage pipes [13,39]. The choice of one of these materials is determined by economic factors. It is worth noting that while PVC and HDPE, as raw materials, have different physical properties, it is possible to make durable pipes of good quality from either materials. In the United Kingdom, drains are made of HDPE and a minority of PP, while in the rest Europe, corrugated drains are mainly made of PVC. In the USA and Canada, most drainpipes are made of HDPE, mainly because of the low price of the raw material [8].

## 3. Evolution of Envelope Technology

Soil particles entering and clogging subsurface drains have been a problem since the beginnings of tile drainage. Clogging of subsurface tile drains with silt and sediment was reported as early as 1834 and probably occurred much earlier. Such deposition in the drains not only reduces the efficiency of the drainage system but also shortens its effective life. The primary reasons of placing envelope material around subsurface drainpipes are usually identified as: (a) to limit the amount of soil particles that may pass through and settle within drainpipes, reducing their capacity; (b) to increase the effectively high permeable radius around drainpipes; (c) to provide suitable bedding for the drainpipe; (d) to increase soil stabilization on which the drainpipe is laid [40]. Envelope materials play an important role towards successful drainage of any waterlogged soil.

*3.1. Terminology Definition*

There is no unified definition of "drain envelope" or simply, "envelope" associated with subsurface drainage that is commonly accepted by the scientific community. A variety of terms exist for envelopes, reflecting the purpose and method of application. Specifically, in the literature, "drain envelope" or "envelope" seems to be a mixture of descriptions of the materials used under different names and functions (filter, hydraulic, mechanical, bedding) [39]. For example, Dieleman and Trafford [41] used the terms "envelope", "filter", and "surround", to distinguish between types of envelopes based on their principal function. Youngs [42] used the terms "filter around drains" and "filter surround". USDA [43] distinguished the terms "drain envelope", "hydraulic envelope", and "filter envelope". Wright and Sands [44] used the term "drain envelope or sock". Luthin [2] used the term "tile filter or envelope" and Scherer et al. [45] used the term "sock envelope". The term "envelope" has also been used as a generic name for any artificial material placed on or around a drain to improve its functioning without specifying the reason for its use. For example, Willardson [40] defined the drain envelope as a porous material placed around subsurface drains. According to Cavelaars et al. [46], common terms are filter, cover material and permeable fill. According to the Irrigation Dictionary of ICID (International Commission of Irrigation and Drainage—Term No. 5097),

the "envelope" is a permeable material that is placed around the drainpipe to prevent fine particles entering into drain, e.g., synthetic, granular, or organic materials [47]. However, in our times, it seems that the prevailing term is "drain envelope" or simply, "envelope".

In early stages of development, design criteria for drain envelopes used the filter concept as a basis for research; consequently, the term "drain envelope" is often mistakenly referenced to as "drain filter". However, drain envelopes and filters are two different means used to solve different problems. Drain envelopes are used: (a) to protect drainpipes against the soil particles entering where they may settle and eventually clog the pipe; (b) to facilitate the flow of water into drainpipes by creating a more permeable zone around drains. With a small modification concerning the definition of "filter material" of United States Department of Agriculture (USDA) [43], we can say that "drain filter" is a layer or combination of layers of pervious materials, designed and installed in such a way as to facilitate the water movement and simultaneously, to prevent the movement of soil particles in the flowing water. It is worthwhile to note: (a) drain envelopes are not filters and (b) while filters become clogged gradually with time, drain envelopes do not.

During the past several decades, several definitions have been used to describe drain envelopes that include its functions relative to soil and trench hydraulics, trench stability and as bedding materials for drainpipes. Table 1 shows various definitions which presented in the international literature and the corresponding citation.

**Table 1.** Various definitions, which presented in the international literature and the corresponding citation.

| Citation | Definition |
|---|---|
| USDA-SCS [48] | Permeable materials placed around the drains for the purposes of improving flow conditions in the area immediately surrounding the drain, and to improve pipe bedding conditions. |
| Ritzema [49] | Material placed around pipe drains to serve one or a combination of the following functions: (i) to prevent the movement of soil particles into the drain; (ii) to lower entrance resistances in the immediate vicinity of the drain openings by providing material that is more permeable than the surrounding soil; (iii) to provide suitable bedding for the drain; (iv) to stabilize the soil material on which the drain is being laid. |
| Cavelaars et al. [46] | Material placed around pipe drains to perform one or more of the following functions, namely (i) filter function, i.e., to prevent or restrict soil particles from entering the pipe where they may settle and eventually clog the pipe; (ii) hydraulic function, i.e., to constitute a medium of good permeability around the pipe and thus reduce entrance resistance; and (iii) bedding function, i.e., to provide all-round support to the pipe in order to prevent damage due to the soil load. |
| Stuyt and Willardson [50] | Porous material placed around a subsurface drain to protect the drain from sedimentation and improve its hydraulic performance. |
| Vlotman et al. [39] | Porous material placed around a perforated pipe drain to perform one or more of the following functions: (i) filter function namely to provide mechanical support or restraint of the soil, at the drain interface with the soil, to prevent or limit the movement of soil particles into the drainpipe where they may settle and eventually clog the pipe; (ii) hydraulic function namely to provide a porous medium of relatively high permeability around the pipe to reduce entrance resistance at or near the drain openings; (iii) mechanical function namely to provide passive mechanical support to the pipe in order to prevent excess deflection and damage to the pipe due to soil load; and (iv) bedding function namely to provide a stable base to support the pipe in order to prevent vertical displacement due to soil load during and after construction. The functions (iii) and (iv) can only be achieved with gravel and sand envelopes. |

| | |
|---|---|
| Wright and Sands [44] | Material placed around a drainpipe to provide either hydraulic function, which facilitates flow into the drain, or barrier function, which prevents certain sized soil particles from entering the drain. |
| USDA-NRCS [51] | Generic term that includes any type of material placed on or around a subsurface drain for one or more of the following reasons: (i) to stabilize the soil structure of the surrounding soil material, more specifically a filter envelope; (ii) to improve flow conditions in the immediate vicinity of the drain, more specifically a hydraulic envelope; (iii) to provide structural bedding for the drain, also referred to as bedding. |
| Nijland et al. [13] | Porous material placed around a perforated drainpipe: (a) to prevent or restrict soil particles from entering the drainpipe where they may settle and eventually clog the pipe (filter function); (b) to provide a porous medium of relatively high permeability around the pipe to reduce entrance resistance (hydraulic function). |
| Stuyt et al. [8] | Porous material placed around a subsurface drain, to protect the drain from sedimentation and improve its hydraulic performance. |

Hereafter, we adopt the aforementioned definition of Vlotman et al. [39].

*3.2. Evolution of Materials Used Around Drainpipes from the Beginnings of Tile Drainage Until Our Time*

In an early drainage text, Stephens [52] cited: "*In clayey or mixed soils, where the water enters the drain at different depths, stones, gravel, or smithy danders, are the only materials that can be used with advantage; in any case, however where tiles are used, the space above them must be filled to the surface of the ground with some porous material, otherwise the drains will be useless, and the undertaking will prove a complete failure*". A couple of decades later in another drainage text, French [26] stated: "*Obstruction by sand or silt: Probably, more drains are rendered worthless, by being filled up with earthy matter, which passes with water through the joints of the tiles, than by every other cause*". In addition, Elliott [53] stated: "*the ends of the tiles ("circular clay pipes, 1 ft long") should be placed close together, in order to prevent the soil from entering, yet not so close as to prevent the entrance of water*".

Therefore, subsurface drainage designers were obliged to invent and apply means and methods of preventing entrance of soil materials in subsurface drains. Some of these specify a variety of methods and means of preventing drain sedimentation. Vlotman et al. [39] pointed out, since 1859, engineers have employed various means to protect drain openings from the entry of sediment, with varying degrees of success. Until today, different materials have been used to prevent drains from clogging, such as topsoil, sod, building paper, strips of tin, hay, straw, corncobs, cloth and burlap, leather, wood chips, sand, gravel, and other more modern materials. French [26] recommended two methods of preventing drain sedimentation, namely (a) the use of double-walled or sheathed drains with collars; (b) the surround of the drain with clean, fine gravel. Elliott [24] cited: "*much difficulty is encountered with sand entering the tile*". He recommended gravel as the best material to prevent this and he believed that it should be used, if possible. Specifically, he considered the construction of a permanent filter from gravel packed around the tile, which admits water, but prevents silt from entering the drains. Brown [54] reported the use of gravel along the sides of wooden box drains in an experiment in 1906 and concluded that unless the drain openings were protected by gravel and sand, covered drains could not be used in certain soils. Hart [55] recommended graded gravel, ranging in size from sand to pebbles one inch in diameter as an excellent filter. He also recommended a porous fabric covering for the tile joints such as burlap or cheesecloth for quicksand conditions. Murphy [56] recommended that tile should be surrounded by porous material, such as gravel or broken rock, that will not erode or permit fine sediment being carried through it. Powers and Teeter [20] recommend in fine textured soil, after the tiles are blinded in or partly hidden with loose earth, covering with a layer of rushes, straw, or sods, to facilitate the entrance of water into the drains. As they cited "*rushes or sod placed over the tiles will be very slow to decay and will prevent the joints from filling up with clay or silt*".

As noted above, the problem of clogging in subsurface tile drains was known at least as early as the 1830s and probably earlier. In the beginning, the necessary envelope material around the tiles was

coming from locally available materials, like sand, stones, gravel, wood fibers or straw. Over the years, envelope materials have included almost all permeable porous materials that are readily and economically available in large quantities and used as drain envelope material to protect subsurface drainpipes and improve or maintain performance.

According to the composition of the materials used, envelopes can be divided into three categories, namely mineral, organic and synthetic envelopes. Mineral envelopes consist of coarse sand, fine gravel, crushed slag and crushed stone. Organic envelopes include bamboo, cedar leaf, cereal straw, chaff, chopped flax, coconut fiber, corncobs, flax stems, flax straw, heather bushes, grass sod, peat litter, reeds, rice straw, sawdust and wood chips. Synthetic envelopes are usually prewrapped loose fibers or granules and geotextile fabrics specifically manufactured for use in drainage and soil stabilization. These synthetic fabrics are made of polyester, polypropylene, polyamide, polystyrene, and nylon [8,46,50].

The evolution of materials used around drainpipes follows three generations of envelopes.

(a) The first-generation envelope material consisted of gravel, broken shells or loose organic materials. According to Juusela [57], the various organic materials used included peat litter, corn, chips, reeds, heather and grass sod. From these, the two most common organic materials are straw and sawdust. Generally, these materials were locally derived. In arid areas, the technique of using gravel envelopes was later further developed to such a degree that effective gravel envelopes can be designed for most soils [58]. Such design criteria for mineral granular envelopes evolved in the United States [40]. Alternative envelope materials were usually composed of organic fibers, such as those found in crop residues. Peat envelopes, for example, were applied successfully for many years and were traditionally used in areas where gravel was expensive [8].

(b) The second-generation envelope material consisted of cover materials in strip form, gradually replacing loose organic materials. A significant advantage of this roll strip is that it can be carried on a trencher and rolled out over the pipe as it is installed. The first materials produced in strip-form were fibrous peat, flax straw and coconut fibers. In addition, strips of glass fiber sheet were used in the 1960s, which were affordable and easy to handle. In 1962, fibrous organic envelopes developed in Europe could be wrapped around corrugated pipes before their installation. This enabled installation of pipe and envelope as a composite product, namely a wrapped drain that reduced installation costs by roughly 50%. For this reason, their use became widespread. However, they showed a proneness to microbiological decomposition, a key disadvantage [8].

(c) The third-generation envelope material, consisting of synthetic fabrics (e.g., geotextiles or loose synthetic fiber wrappings) wrapped around the drainpipe, has rapidly gained popularity. Most loose synthetic fiber wrappings are manufactured from recycled material, such as polypropylene waste fibers from the carpet industry. During the past 45 years, rapid technological developments have occurred in drainage techniques and materials. Today, specialized machines can pre-wrap sheet and loose fiber envelopes around the drainpipes. Such pre-wrapping completed in the factory and not in the field, ensured a better quality envelope-pipe system and greater quality control at installation [59].

## 4. Evolution of Installation Technology

The classical procedure to install a pipe drainage system is: (a) marking the alignments and levels; (b) excavating the ditches at the required depth and gradient; (c) placing the pipes in the ditches, with or without cover or envelope material; (d) backfilling the ditches with the excavated soil.

This type of construction system requires suitable excavation equipment and construction materials. For the establishment of a proper installation of pipes, various machines and instruments were devised. This equipment can be divided into dredging equipment, excavation machines, equipment for construction control, ancillary machinery and equipment, maintenance machinery and equipment for inspection of drains [60].

*4.1. Early Stages*

For many centuries, manual labor was used to install drainage systems, namely digging the trench to a suitable depth and grade, laying each segment of tile in place at the bottom of the trench, and then, backfilling the trench. The tools required for digging ditches by hand are simple and not numerous. They consist, mainly, of shovels, spades, scoops and picks. In addition, several special tools have been developed for manual installation of the drains, such as the laying hook, soil pincer, hatchet pick, among others. Figure 6 shows a set of drainline digging and installation hand tools.

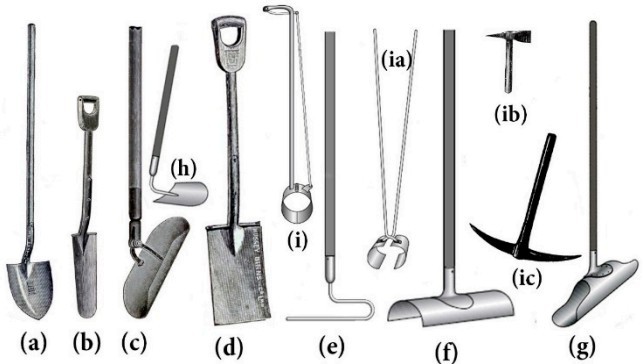

**Figure 6.** Digging and installation tools used in constructing tile drains by hand: (**a**) Ordinary Shovel [61]; (**b**) Bottoming Spade [29]; (**c**) Bottoming Scoop [29]; (**d**) Ordinary Spade [61]; (**e**) Laying Hook [62]; (**f**) Correction Hook [13]*; (**g**) Drain Scoop [13]*; (**h**) Hoe [13]*; (**i**) Pipe tongs [13]*; (**ia**) Soil pincer [13]*; (**ib**) Hatchet Pick [25]; (**ic**) Hand Pick [63]. (Conception, synthesis and processing have been made by Stavros Yannopoulos) (* by permission of Alterra-ILRI Publisher).

Figure 7 shows a laborer digging a ditch and Figure 8 a laborer laying tiles. In Figure 7, note the cord on the left side of the trench used as a guide to help keep the trench straight.

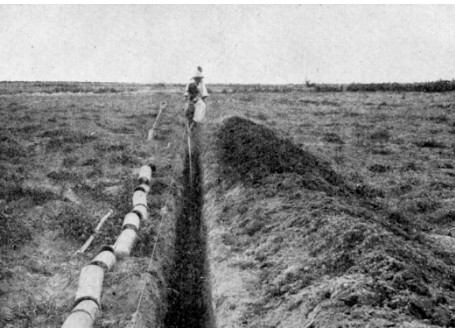

**Figure 7.** Laborer digging a ditch [64].

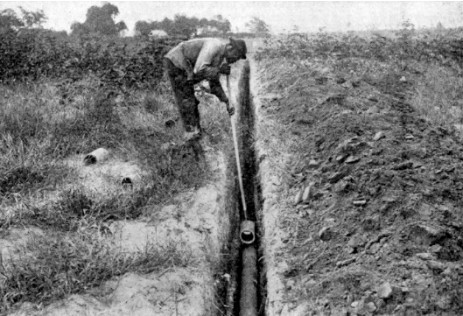

**Figure 8.** Laborer laying tile [64].

After installation and inspection, drain tiles were blinded by workmen standing beside the ditch, shoving excavated dirt from the sides of the ditch with a spade [20]. Refilling the trench was accomplished either by hand with a spade or/and horse drawn (i.e., plow) or power operated (i.e., V

Drag, Grader, etc.,) implements [16,63]. Other tools used for filling ditches were V-Crowders, which are shod, light road-drags, and light road-graders [20].

Figures 9 and 10 illustrate ditch filling with a plow and teams and a road grader, respectively, while Figures 11 and 12 present filling up a closed drain with a dragline excavator and power-operated backfiller filling a large tile trench, respectively.

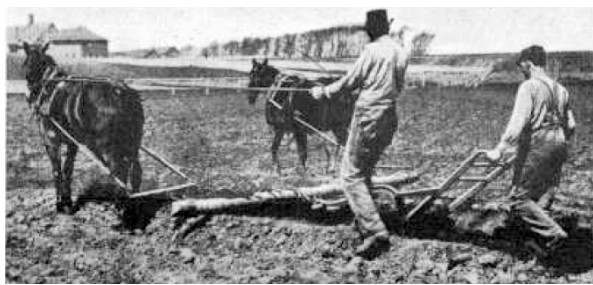

**Figure 9.** Filling up the ditch with a plow [19].

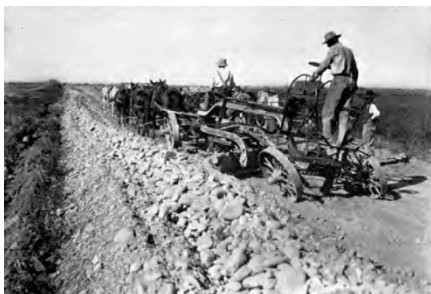

**Figure 10.** Filling up closed drain with teams and road grader [56].

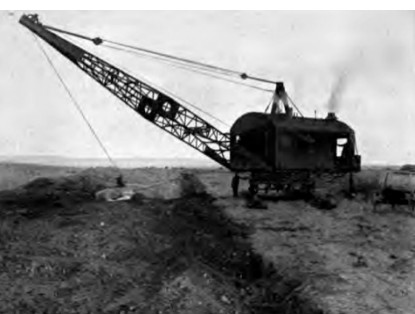

**Figure 11.** Filling up closed drain with dragline excavator [56].

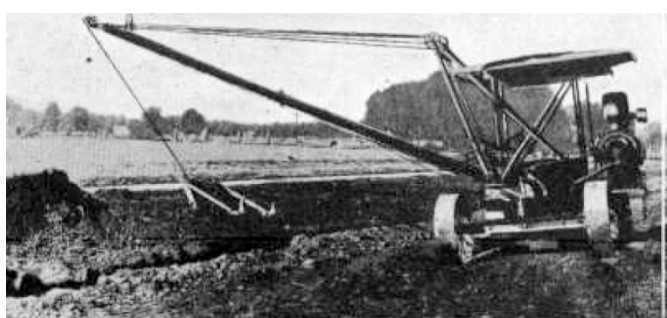

**Figure 12.** Power-operated backfiller filling a large tile ditch [65].

Manual trench digging methods limited trench depths to about 1.5 m (5 ft) deep and the bottom seldom exceeded about 1.2 m (4 ft) [66]. It remains challenging to place hand-dug drainage systems on the desired levels and grades for distances greater than about 500 m. In early drainage texts, details of the installation procedure are described [3,19,24,26,32,52,56].

Today, installation of drainpipes by manual labor is rare in many countries. Nevertheless, hand-dug trenches may still be of importance in areas where machines are not readily available or where small drainage works are to be implemented or labor is abundant and economical. Consequently, this method of drainage system installation by hand is still practiced sporadically.

*4.2. Evolution of Installation Technology Until Around the Mid-20th Century*

The process of manual installation of the drainpipes is time consuming, laborious and expensive, and depends entirely on the cost and availability of labor. In general, manual installation of tiles is much more expensive than machine installation. For example, drain installation by hand required 230 to 300 man hours per 1000 m drain [67]. Of course, modern mechanical installation is generally more cost effective and usually results in a better quality installation [13]. Nevertheless, when groundwater or desired drain depths are shallow (<1m), manual installation can be an alternative.

However, apart from the installation cost, the time consuming process and the scarcity of qualified hand tilers, the need remained for improved methods of constructing large mainline trenches or outlet ditches, which were important factors in the spread and efficiency of drainage.

Thus, after centuries of drainage systems' installation by hand, the invention of the trenching machines and the steam engines caused a revolution in the practice of land drainage installation in the late 19th century.

Yarnell [65] pointed out, "*the more extensive use of tile-trenching machinery has been brought about by the rising prices and increasing scarcity of labor and the rapid extension of tile drainage for farm lands to increase crop production*". Jones [68], concerning the movement of the machines, considered that tile-trenching machinery may be divided into two general classes, namely: (a) horse-drawn ditching plows; (b) power-operated trenching machines. In reality, horse- or ox-drawn plows were already in use for digging ditches. These methods were economical only for ditches up to 1.5 m (5 ft) deep with bottom widths seldom exceeding about 1.2 m (4 ft). However, usually these ditches did not provide the drainage desired [14].

Around 1855, the Pratt and Brothers Co. at Canandaigua, New York, manufactured a sort of horse-drawn scraper, called Pratt's Ditch Digger (Figure 13), with which a trench could be dug by means of a wheel. This first ditch-digging machine was called "Buckeye" [69]. Pratt's machine (Figure 13) is worked by one or two horses, or by steam or water power, as was available locally [70]. The New York State Agricultural Society in the 1855 annual report considered Pratt's Ditch Digger as "*the most valuable machine*" [70].

By 1880, several types of horse powered machines were developed, such as the Johnson Tile Ditcher, the Hickok and the Rennie elevator Ditchers, etc. However, these machines had a serious disadvantage, namely they all required a number of passes over the trench to achieve the required depth, while the trench bottom had to be finished by hand. Later, single-pass machines were developed, though still powered by horses. One of these was Heath's Ditching Machine (Figure 14), which operated on a wooden track, powered by one horse on a sweep. Additionally, there were Paul's Ditching Machine (Figure 15), Fowler's Drainage Plow (Figure 16), among others.

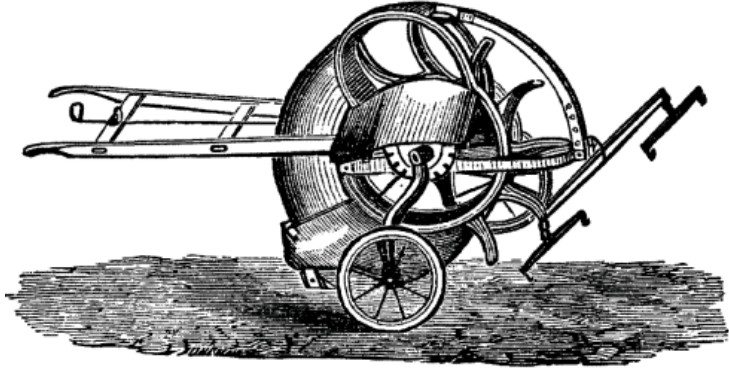

**Figure 13.** Pratt's Ditch Digger [26].

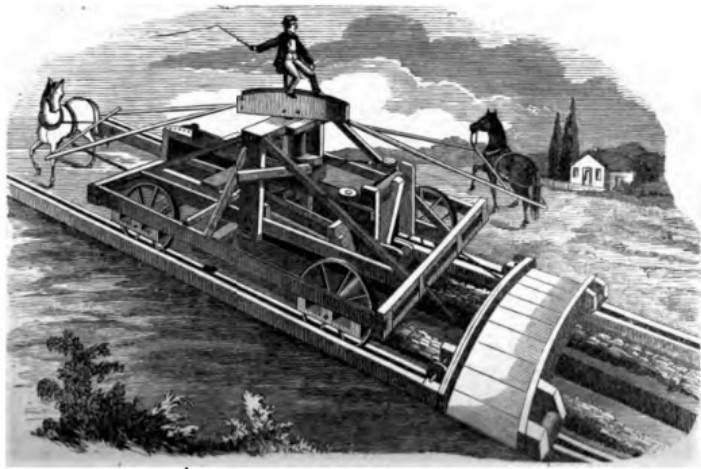

**Figure 14.** Heath's Ditching Machine [71].

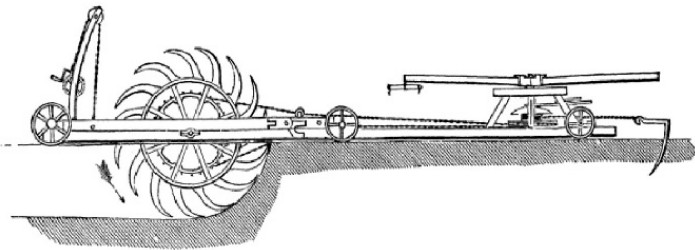

**Figure 15.** Paul's Ditching Machine [26].

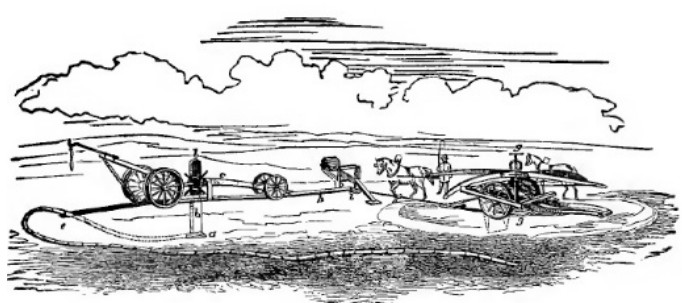

**Figure 16.** Fowler's Drainage Plow [26].

In particular, the single-pass machine is made to pass back and forth over the ditch, each time increasing the depth until the design depth is reached. One such machine is Johnson's Tile Ditcher (Figure 17), which is drawn by eight horses. However, there were other machines that completed the ditch as the machine advances, requiring only one pass over the ground. One such machine is the Blickensederfer Tile Drain Ditching Machine (Figure 18) that was powered by a single horse, one man and one boy [30].

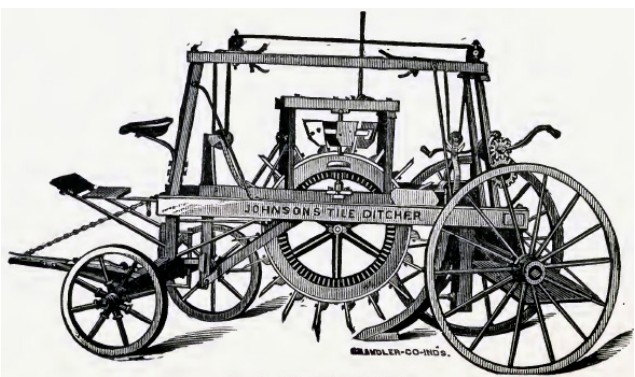

**Figure 17.** Johnson's Tile Ditcher [30].

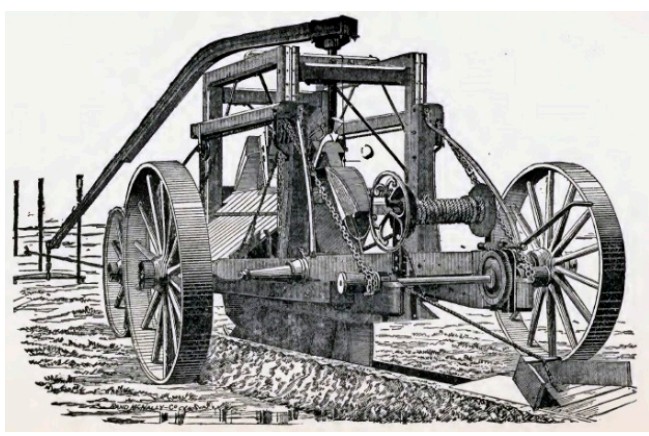

**Figure 18.** Blickensederfer Tile Drain Ditching Machine [30].

Commenting on the development of excavating machinery up to 1915, Yarnell [72] stated: "*The increasing demand for suitable excavating machinery has engaged the attention of many men of mechanical bent, and the result has been the invention of modern types of machinery, the development of which has been rapid. By the use of modern machinery the cost of drainage work has been so reduced as now seldom to afford valid excuse for failure to drain*". In reality, machines driven by steam engines for trench excavation of trenches began about 1890 [73], followed by the appearance of dragline excavators in 1906 in the United States [9]. In 1883, the steam-powered Plumb Ditching Machine was developed. For many years, such machines were widely used [14,66]. In 1892, the steam-driven ditcher invented by James B. Hill (1856–1945) in Ohio, USA, was well-known as the Buckeye Steam Traction Ditcher (Figure 19a,b). Broughton and Fouss [74] characterized this advance as "*a welcome era replacing difficult hand digging of trenches by machine work*". This ditcher was the forerunner for traction ditchers used worldwide [75] and today's high speed trenchers and laser-controlled drain plows [14]. A description of the machine and its capabilities as well as details of its operation are presented by Perkins [76]. Later models of the machine were larger and by 1908 used gasoline engines, then by 1920, they were diesel-fueled [75].

Yarnell [72] noted that "*although many delays and difficulties were encountered in the early stages of development, the cost of excavation by machinery was soon reduced far below that by hand labor*". In addition, he believed "*this period marks an epoch in the progress of drainage*" in the USA.

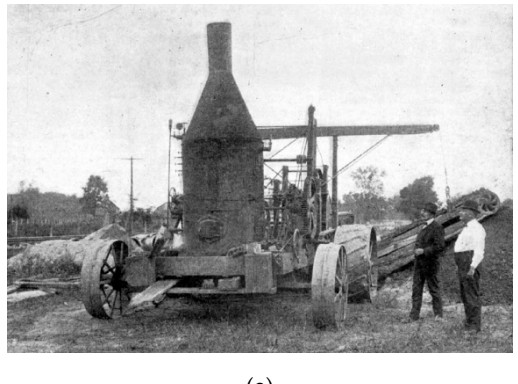 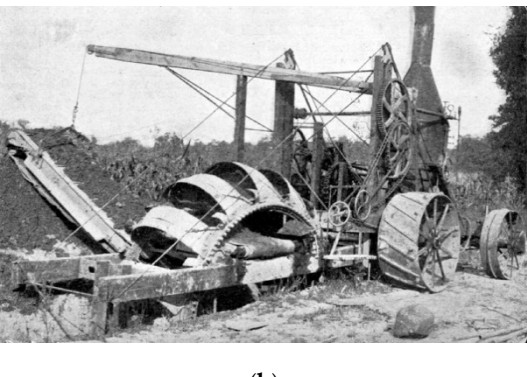

(**a**)     (**b**)

**Figure 19**. The steam-driven Buckeye Steam Traction Ditcher: **a**) Rear view, (**b**) View showing the Buckets of the machine [76]

Yarnell [65] and Beach [77] classified tile-trenching machines into four general classes, namely (1) plows (Figure 20a,b) and scoops; (2) wheel excavators, as the Cleveland Trencher, the Parsons Trench Liner, the Buckeye Farm Drainage Ditcher, etc.; (3) endless chain excavators, as the Badger Trench Excavator, the Barber Greene Standard Ditcher, the Jeep-A-Trench, etc.; (4) scraper excavators, as the Schield Bantam Trench Hoe, etc. The most common of these were the wheel excavators and the endless chain excavators.

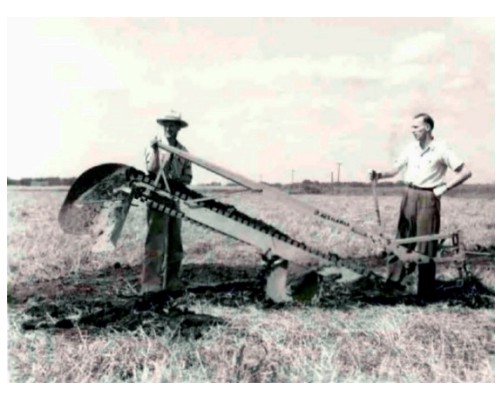 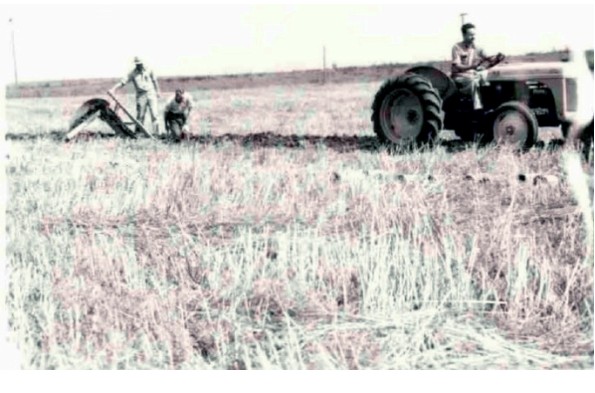

(**a**)     (**b**)

**Figure 20.** (**a**) Aashmar's Draining Plow (**b**) In use [77].

With time, manufacturers constantly improved the trenching machines, adding many new features to increase their capabilities, making them more powerful and easier to operate. In addition, several types of these machines were produced in various sizes and with a wide range of capacities. It is worth noting that mechanized trenchers were part of standard drainage installation equipment for decades, and many of them remain in use today.

*4.3. Evolution of Installation Technology Until the Present*

According to Yarnell [72], "*perhaps the first successful use of power machinery in drainage work was on a project in Illinois in 1882 when a floating dredge was used for digging the channels*". The need for drainage across the States motivated drainage engineers and machine fabricators to build a wide range of trenchers, pipe laying plows and power shovels capable of efficient, effective and economical pipe installation. The invention of the combustion engine in the 20th century led to the development of new machines. As Ritzema et al. [59] noted "*in the 20th century, the appearance of fuel engines led to the development of high speed installation techniques of subsurface drains with trenching or trenchless machines*". The mechanical installation of subsurface drainage was introduced in Canada in 1902 [78], in Egypt by 1960 [79], in the Netherlands by 1954 [67], in Denmark by 1971 [80], in France by 1974 [69], etc. In general, the mechanical installation of subsurface drainage was introduced in Europe in

the 1950s [67]. In fact, mechanized installation equipment developed rapidly from the 1940s onwards [13]. Around 1960, mechanical installation became widespread [8]. First, the so called trencher machines were introduced, followed in the late 1960s by the introduction of trenchless machines [13,46,74].

Nowadays, drainage pipes are installed by drainage machines, from which the most common types can be classified into two major categories [46]: trenchers and trenchless machines. Trenchers dig a trench in which the pipe is laid, whereas trenchless machines merely lift the soil while the pipe is being installed. Specifically, a trencher digs the trench with its bottom at the desired depth and required grade and simultaneously lays the drainage tile on the bottom of the trench. After laying the pipe, the trench is backfilled by hand or by motorized equipment. Trenchers are manufactured in various types and sizes with a large range of abilities, in particular: (a) to install pipes to a depth of about 3 m, in hard or stony soil, in unstable subsoils and or under the groundwater level; (b) to make trenches up to 0.50–0.60 m wide; (c) work in soils with hard layers [13].

There are two basic types of trenching machines, namely the wheel type (Figure 21a) and the ladder or chain type (Figure 21b) [81].

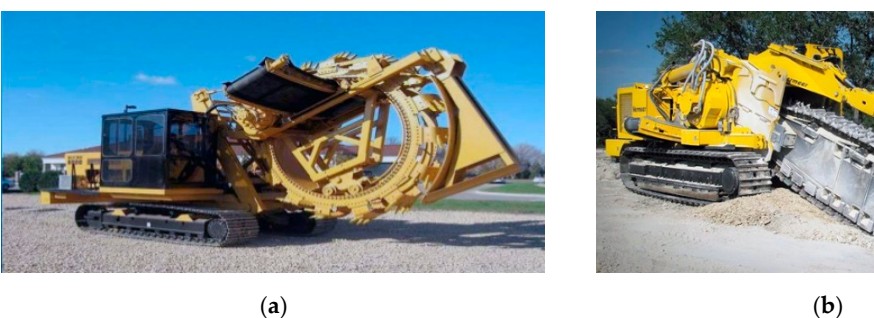

| (**a**) | (**b**) |

**Figure 21.** Trencher machines: (**a**) wheel type [82]; (**b**) ladder or chain type [83].

The trenchless drainage machine is also called a "drain plow" [67]. Perhaps the concept of "plowing-in" subsurface drainage conduits dates to at least the mid-1850s [81], as in 1860, French [26] reported the Fowler's Drain Plow (Figure 16). However, a further literature search reveals that several draining plows were invented and patented in England in 1834 and thereafter [84], but were designed to make the surface covered or underground drains of only a few inches in depth. Even earlier, however, Curven et al. [85] notes that Mr. Adam Scott had invented an instrument, called the "Mole Plow" that he claimed was valuable towards making underground drains. Mr. Scott presented this machine at the Society for the Encouragement of Arts, Manufactures, and Commerce, Instituted at London. Various draining plows are described and commented on by Klippart [3], such as the Adam Scott mole plow, the Cole and Wall mole plow, the Gopher plow of Illinois, the Rowland and Forbis mole plow (also known as the Witherow plow) and many others. Figure 22 shows the Adam Scott plow, which is the pioneer mole plow according to Klippart [3].

Trenchless drainage only became practical in the mid-1960s with the development and introduction of corrugated-wall plastic tubing [13,74,81]. Trenchless drainage machines can lay drainage tiles at a higher speed than trenchers and do not require any backfilling, since there is no excavation. Nevertheless, their use can be more restrictive than the trencher machines when it comes to the depth and size of drainage tiles due to capacity limitations, as they require more power for the same installation depth than trenchers [13]. Ritzema [86] pointed out "*the advantages of trenchless drainage decrease rapidly with greater drain depth and heavier soils*". Moreover, the trenchless machines have more restrictions than trenchers, given the fact the maximum installation depth is about 1.8 m and only corrugated plastic pipes and pre-wrapped envelopes can be used. Corrugated plastic pipes are the only feasible pipes for trenchless machines. The V-plow can handle a maximum outside pipe diameter, including the envelope, of 0.10–0.125 m, while the vertical plow can handle much larger diameters [46].

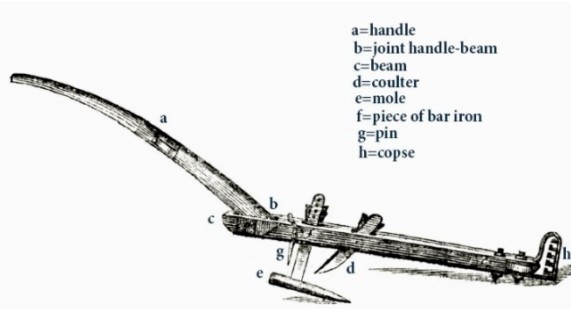

**Figure 22.** The pioneer mole plow of Adam Scott [85] with modifications by Stavros Yannopoulos.

There are two main types of trenchless drainage machines, namely the vertical plow (Figure 23a,b,c) and the V-plow (Figure 24a,b,c).

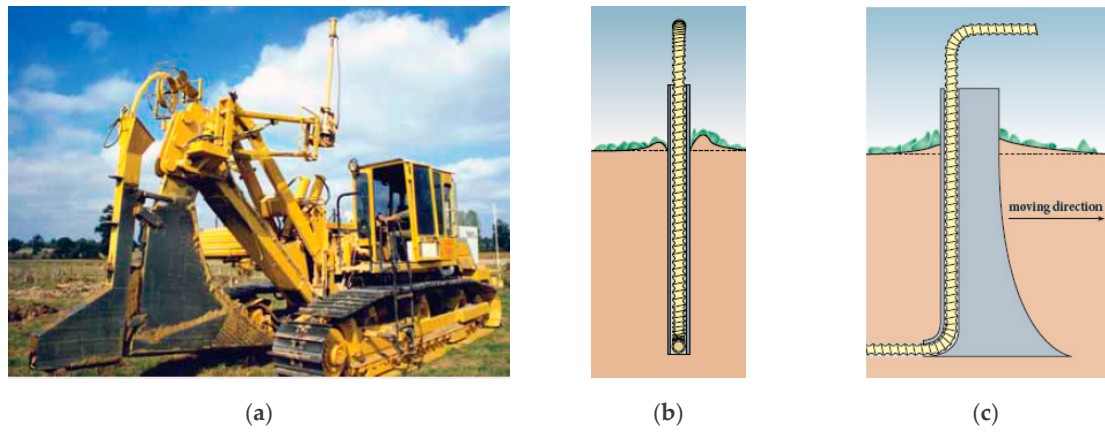

(**a**)            (**b**)            (**c**)

**Figure 23.** (**a**) Vertical plow type trenchless drainage machine; (**b**) Rear view; (**c**) Side view [13, with permission of Alterra-ILRI Publisher]).

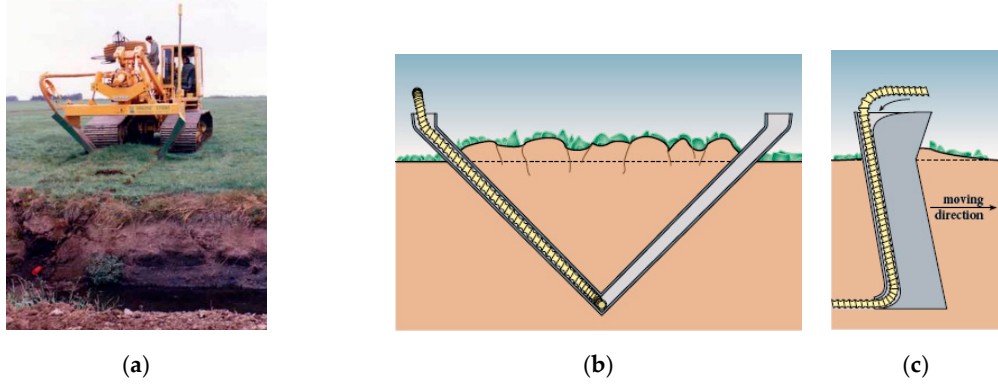

(**a**)            (**b**)            (**c**)

**Figure 24.** (**a**) V-plow type trenchless drainage machine; (**b**) Rear view; (**c**) Side view [13, with permission of Alterra-ILRI Publisher]).

Historically, the digging machine operator visually controlled the depth and grade of the drains. In particular, the operator hydraulically raised or lowered the digging mechanism to bring a sighting bar in line with crossbars on targets aligned across the field. However, these machines were slow moving. Modern high-speed drainage plow equipment use laser-controlled hydraulics to ensure accurate drain installation depth and grade. Similarly, the laser control system is combined with a GPS (Global Positioning System) unit to guide exact spatial placement of the drainage system with respect to its outlet and other infrastructure. It is worth noting the cost of laser equipment does not exceed 10%, at most, of the cost of the drainage machine [13].

Well-managed trencher machines can achieve drainline installation speeds of about 2 km/h. In reality, the actual installation speed per day must take into account the equipment transport, maintenance, preparation and setup times. Trenchless drainage machines can install 4 km/d in a logistically well-organized environment [13].

## 5. Drainage System Design/Installation Effects on Drain Water Quality

The risk posed by discharge of drain water containing nutrients, salts and other contaminants to the ecology of receiving waters, such as streams and wetlands, is of growing concern and must now be addressed by those responsible for management of agricultural production, whether irrigated or otherwise. For example, in parts of the San Joaquin Valley of California, farmers dispose low quality drain water by evaporation from constructed collection ponds, or reapply the saline drainage water to successively more salt-tolerant crops until the remaining collected brine is used in solar energy generation ponds. The drain water evaporation ponds have a limited future associated with the fate of the accumulating salts precipitating as the water evaporates (Tanji and Grismer 1987) [87]. On the other hand, in more humid regions, agricultural drain water can safely pass through wetlands with minimal adverse impacts, while simultaneously improving the chemical and biological quality of the water. However, more work is needed to define acceptable drain water quality criteria for wetland and stream disposal.

Subsurface drainage systems collect the root-zone drainage water that presumably reflects the water quality (salinity, mineral dissolution, excess applied nutrients, herbicides and pesticides) of the root zone leachate and depends, in part, on the relative efficacy of the drainage system in collecting this leachate across the soil profile width between drain laterals. Drainline spacing and depth together with varying soil permeability control the water flow paths within and below the root zone which then influences the water quality of the collected drainage water [88]. Drain water salts' load–flow relationships are a measure of the drainage system capacity and efficacy to extract salt and reduce the salinity of the root zone while also indicating something of the possible nutrient and other pollutants captured by the drainage system. Theoretically, when the salt concentration is constant, there is a direct, usually linear, relationship between salt load and drain discharge or flow rate; that is, as the flow rate increases, there is an increase in the salt load, partly due to the increase in flow. Grismer et al. [7] noted that the previous investigators, Frank E. Robinson and James N. Luthin, found that when originally installed in the 1960s, the bituminous fiber and clay tile drainage systems yielded a decreasing salt load with increasing flow, as a result of "*a unique combination of hydraulic conductivities along each tile line*" such that "*areas of high hydraulic conductivity tended to dilute the salinity in the areas of low hydraulic conductivity*". However, later, Grismer et al. [7] found a linearly increasing relationship between salt load and drain flows as expected, though a newly installed drainage system yielded a roughly 20% smaller load for the same drain flowrate as compared to that installed some 20 years prior. Grismer [88] and Guitjens et al. [89] underscored that "*as mechanisms of in-situ water mixing, salt mobilization, and solute transport are better understood, drainage designs should include criteria that make it possible for management to manipulate the quantity and quality of drainage discharge*".

Ideally, mechanistic models should give reasonable estimates of the mass of water, salt, and the individual salt components involved in subsurface water movement. When water flows downward through the vadose zone to the water table and becomes groundwater, the saturated conditions may change the chemical processes. Even process-oriented models of the vadose and saturated groundwater zones are necessarily based on simplifying assumptions, such as chemical equilibrium, which may not occur in reality. Accurate data on vadose zone and aquifer minerals, pH and redox status, and the heterogeneous nature of the media are difficult to acquire. Grismer [88] used steady state and transient numerical simulations to model the effects of depth and spacing of the drains on drain discharge and quality in terms of salt concentration and load. Under conditions of increasing salinity with soil depth, results of the steady state model yielded increased drain water salinity as either drain spacing or depth increased, but the increase with drain depth was more pronounced. Flow paths and groundwater displacement remained somewhat invariant in transient flow. The deeper drains and wider spacing had greater flow, salt concentration, and salt load. Regardless, from

a water management perspective, decreasing drainage flows through improved irrigation application efficiency will likely reduce contaminant loads to receiving waters accepting drainage discharge, whether surface or groundwater.

## 6. Alternative Drainage Technique to Control Waterlogging and Salinity—Biodrainage

The conventional drainage technologies (surface drainage and subsurface drainage) have two basic drawbacks, namely, they are costly and they generate drainage effluents, which will have to be either carefully reused or safely disposed of. It is worth noting that the management of drainage effluents has become an important issue around the world. Moreover, lack of finance and inadequate maintenance often impede their installation and sustainability, respectively. The aforementioned reasons prompted the search for alternative approaches to maintaining sustainable agriculture in the long run. These alternatives must be effective, affordable, socially acceptable, environmentally friendly and not cause degradation of natural land and water resources.

The technology known as biodrainage is a fairly new technology that is environmentally friendly and of less cost.

The term biodrainage first time documented by Gafni [90], however, earlier, Heuperman [91] used term biopumping to describe the use of plants for water table control. It has also included the term of biodisposal, which refers to the use of plants for final disposal of excess drainage waters. Biodrainage could be used in irrigation schemes to control the rising water table and avoid waterlogging and salinization [92].

Biodrainage processes can be classified based on land use context as: (a) Dryland/rainfed systems: (i) recharge control, (ii) groundwater flow interception and (iii) discharge enhancement; (b) Irrigated systems: (i) water table control, (ii) channel seepage interception, and biodrainage cum conventional drainage systems [92].

Plants, especially trees, with high evapotranspiration rates should be used. The evapotranspiration principle of plants is used in biodrainage treatment to reclaim such problematic areas sustainably (Figure 25). Evergreen broad-leaved species that recorded high evapotranspiration rates could be used for reclamation of waterlogged soils. Sarvade et al. [93] reported that short rotation fast growing tree species, such as *Salix* spp., *Eucalypt* spp., *Acacia* spp., *Albizia* spp., *Terminalia* spp., *Prosopis* spp., and *Populus* spp. were the suitable species for biodrainage technology. By using such trees, the elevated groundwater table of an area could be reduced [92]. Agri-silviculture, agri-horti-silviculture, silvi-pasture, multipurpose woodlots, strip plantation and boundary plantations were widely used for reclamation of saline-waterlogged conditions of India. In an agri-silviculture system, *Eucalypt* spp. based agro-forestry systems are widely used for reclamation of waterlogged areas as compared to other woody plant-based systems.

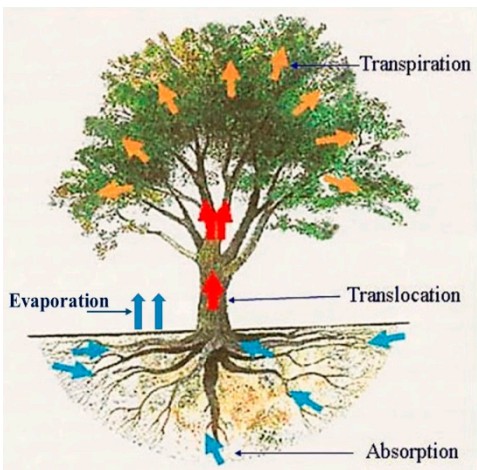

**Figure 25.** Concept of biodrainage (modified from Ram et al. [94]).

In India, highly evapotranspirating tree species selected to mitigate waterlogging conditions [95]. In a study in Rajasthan in northern India, an average annual rate of evapotranspiration was estimated as 3446 mm (which is about 1.4 Class A pan) from a 25 ha land planed with *Eucalyptus camaldulensis*, *Acacia nilotica*, *Prosopis cineraria* and *Ziziphus* spp. in a density of 1900 trees/ha [96]. Additionally, in another study, where numerous plant systems were tested, an *Eucalypt*-based agri-silviculture system gave good results, with 0.84–0.86 m total drawdown of groundwater table in a three-year period and is a widely used vegetation system in India [93].

One special application of biodrainage technology is the amelioration of waterlogged soils during the initial reclamation or ripening phase of '*new*' land development. Vegetation with a vigorous, deep and extensive root system is used to dry out waterlogged soil profiles. For example, in the Netherlands, land to be reclaimed from the sea is sown with reed, while a few centimeters of water remain [92]. Such seeding accelerates the ripening process.

However, experimental results showed that the biodrainage was highest when the groundwater salinity was lowest. Its magnitude decreased with an increase in salinity of the groundwater (Ram et al. 2008) [94]. In Australia and elsewhere, it is widely accepted that in discharge situations, enhanced evapotranspiration biodrainage sites will eventually succumb to salinity, unless some form of conventional drainage is installed to control salt balance to the vegetation's root zone by removal of saline drainage effluent [97]. In that case, plants with salt tolerance (e.g., *Acacia kempeana* and *Eucalyptus tereticornis*) should be used [98] and/or for the final salt balance control, a disposal system is required [92].

## 7. Conclusions

The world history of water reveals how ideas and practices have spread in different directions at different times in a series of transcultural transmissions back and forth, with additions, modifications and improvements that link humanity as a single water community [99].

Over the last two centuries, there has been considerable progress made by engineers and scientists in the field of drainage. It is worth noting that, since the 1950s, the new types of installation equipment, materials and techniques were revolutionary for the drainage industry and contributed to the improvements of the drainage systems' installation quality and durability. Subsurface drainage methods, materials, technologies, and installation were modernized much more through innovative research and development between 1960 and 1975 than during the previous centuries. In particular, since the 1960s and thereafter, drainage technology has changed rapidly, with enormous improvements in drainage tubing, machinery and methods of installation, drainage envelopes (natural and manufactured), and techniques for quality assurance and control. Additive factors that have influenced and contributed to the progress and spread of land drainage are (a) the improvement of land drainage theory; (b) the invention of a new generation of computers with many possibilities in memory and speed; (c) the development of computer simulation models which use the drainage theory to describe the performance of drainage and related water management systems; (d) the development of new tools such as Geographical Information Systems (GIS) and Computer Aided Design (CADD) to the systematic planning and design of drainage systems. It should be noted that this improvement resulted due to the better comprehension of the fundamental processes and relationships through a long time of research and practice, as well of the improved methods that were applied to the effects on the crop: (a) of the soil salinity and (b) of excessive and deficit soil-water conditions. The aforementioned events in combination with the development and the transformation of agricultural drainage contributed to the expansion of agricultural drainage worldwide. Nowadays, agricultural drainage is considered as an integral part of total water management and not just as surplus excess water and salt removal, as in the past.

By 2050, the world's population will reach 9.7 billion (United Nations), namely 24% higher than it is today (7.8 billion in 2020). In order to feed this population and limit world hunger, food production should double within the next 30 years. Therefore, investment is needed in improved irrigation and drainage practices in existing irrigated areas. This means reclamation of areas equipped with irrigation facilities by using piped subsurface drainage systems. In addition, it means

improved water management, which can relatively cheaply restore these areas to their full production potential. For this purpose, it is estimated that about 10–15 million ha of land must be improved for crop production. In comparison with surface drainage, subsurface drainage is the most advantageous solution for several large areas in the world because the maintenance of suitably constructed pipe systems shows far less problematic behavior compared to the maintenance of other forms of drainage. Clearly, subsurface drainage will be an important tool used to increase agricultural yields. We need significant investments in drainage systems' design, materials and installation today to meet the challenge of feeding a constantly growing world population.

More research is also needed: (a) for the development of alternative drainage techniques of low cost to control waterlogging and salinity of soil, which should be effective, affordable, socially acceptable, environmentally friendly and not cause degradation of natural land and water resources; (b) for the development of new technologies of installation of drainage materials so as to reduce installation cost; (c) for the invention of new drainage materials (pipes, envelopes); (d) for the definition of national standards in particular where needed from the local circumstances; (e) for the adoption of new technologies such as aerial photography with drones and satellite remote sensing application, in order to detect the need for drainage and salinity, among others.

**Author Contributions:** S.I.Y. had the original idea, contributed to the project idea development, prepared the manuscript (writing the original draft, reviewing and editing, collecting and preparing the figures), made data collection of the main text and supervised the research; M.E.G. reviewed and revised the manuscript; K.M.B. reviewed and revised the manuscript; A.N.A. revised and edited the manuscript. All authors have read and agreed to the published version of the manuscript.

**Funding:** This research received no external funding.

**Acknowledgments:** The authors would like to thank: (a) The anonymous reviewers for their relevant comments and suggestions which helped to improve the manuscript; (b) The Alterra-ILRI Publisher for their kindness in granting permission to republish Figures 6(f, g, h, i, ia), 23 and 24.

**Conflicts of Interest:** The authors declare no conflict of interest.

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
