# Peer review of "Evolution of the Materials and Methods Used for Subsurface Drainage of Agricultural Lands from Antiquity to the Present"

_water, doi:10.3390/w12061767_

Round 1

Reviewer 1 Report

Dear authors and editor,

The manuscript, entitled “Evolution of the materials and methods used for subsurface drainage of agricultural lands from antiquity to the present”, conducts a comprehensive review of the history and recent developments related to the materials and installation methods for subsurface drainage systems.  The authors have spent considerable time and efforts to collect and organize the materials and presents the information in a way that is logic and easy to understand.  Such information will be valuable for the students, entry-level researchers and engineers, who would like to obtain a big picture of agricultural drainage systems. 

However, as this manuscript is considered as a journal publication instead of a textbook, there exist some critical concerns before the consideration for publication.  First, the research question and testable hypotheses are vague in this manuscript.  In addition, the review does not summary and discuss the problem and needs at the current subsurface drainage systems. The reviewer also has some concerns regarding the copyright issues. Further comments will be given in the following sections.  Therefore, the reviewer suggests the current version of manuscript is not suitable for publication in Water.  However, the manuscript is still a good fit for a book chapter.

Specific comments:

  1. The logic flow in introduction needs some improvement. The current version does not provide enough materials and discussion for research question and hypothesis.
  2. A critical concern is the right to reproduce materials from other publications. For example, the manuscript employed and quoted many materials, i.e. figures (Fig. 23 & 24) and original content (Ln 150-157 and many others). Have the authors requested to reproduce these materials from the copy-right holders?  If so, please attached the paperwork to grant the rights for reproduction from the publishers.
  3. Although the manuscript provides detailed review of subsurface drainage, the current version does not discuss the existing demand and problems that can be addressed by the development of materials and installation methods. For example, how to select the suitable synthetic envelops (geotextiles) in specific regions with the help of advanced techniques?  In addition, to help reduce the dissolved nutrient export from agricultural drainage systems, how does the installation equipment become compatible with on-site treatment practices (e.g. bioreactors or saturated buffers)?  The absence of such discussion substantially reduces the value of this excellent review.
  4. One of the missing part for drainage system is the trafficability (Ln 43-44). In many areas, machinery to plant and harvest crops require drainage to provide suitable trafficability.
  5. Ln 124-128, “lacking drainage pipes … to the rivers”. As the pipes did not exist in the ancient times, the reviewer suggests removing this part and directly jumping to the first discovery of pipes in Ln 128.
  6. Ln 145-146, “Columella states both … narrow”. Surface drainage is relevant to your study in this manuscript.
  7. Ln 133-157, Please be careful that this section describes the materials of subsurface drain pipes. This paragraph is mostly off the topic.
  8. Ln 158, please remove the irrelevant information regarding surface drainage.
  9. Ln 164-166, repeated with Ln 128.
  10. Ln 183-189, the reviewer can not understand the message to be delivered from this paragraph.
  11. Ln 209, the timeline is very confusing here. As the heading mentioned this section reviewed materials from 1800-1940s, why this type of tiles dated back to 1760?
  12. Ln 223, “…several years later…” please specify the exact year.
  13. Ln 237, “… by some 70%”, please revise this.
  14. Ln 240, “For the next county, … in all counties” The reviewer suggests being really cautious about this statement. Not all the counties use the same material for subsurface drainage.
  15. Ln 242-244, the reviewer can not understand the message from this paragraph.
  16. Ln 245, “… both clay and concrete pipes were the main drainage materials…” this statement conflicts with the statement in Ln 240. Please clarify this confusion.
  17. Ln 263-268, it seems that this paragraph does not add too much information in addition to the sentence in Ln 261-262, “Since the 1980s, …for drainage pipes”.
  18. Ln 309-349, these intensive definitions of envelop seems off the topic of your study.
  19. Ln 341-359, another example of intensive quotes from existing literature. The reviewer defers this issue to the editor.
  20. Ln 417-424, further in-depth review of recent development of synthetic envelop is needed as this is the recent hot-spot in related research.
  21. Ln 607, it might be better to replace “epilogue” with “conclusion” (or similar).
  22. After reading this manuscript for multiple times, the reviewer still thinks the current version is logic, well-organized and carefully-prepared. However, due to the lack of key research question and problems and demand from recent studies, the reviewer suggests substantial revision is needed before the publication. 

Author Response

Response to Reviewer 1 Comments

Manuscript ID: water-792752

Title: Evolution of the materials and methods used for subsurface drainage of agricultural lands from antiquity to the present

Authors:  Stavros Yannopoulos*, Mark E. Grismer, Khaled Bali, Andreas Angelakis

Journal: Water

Dear Reviewer,

we express our gratitude for the excellent revision work you made. The answers to your comments are in what follows.

Point 1:    The manuscript, entitled “Evolution of the materials and methods used for subsurface drainage of agricultural lands from antiquity to the present”, conducts a comprehensive review of the history and recent developments related to the materials and installation methods for subsurface drainage systems. The authors have spent considerable time and efforts to collect and organize the materials and presents the information in a way that is logic and easy to understand. Such information will be valuable for the students, entry-level researchers and engineers, who would like to obtain a big picture of agricultural drainage systems.

Response 1: The authors would like to thank the anonymous reviewer for his time, efforts and constructive comments that helped us to re-consider and to improve our manuscript.

Point 2:    However, as this manuscript is considered as a journal publication instead of a textbook, there exist some critical concerns before the consideration for publication. First, the research question and testable hypotheses are vague in this manuscript. In addition, the review does not summary and discuss the problem and needs at the current subsurface drainage systems. The reviewer also has some concerns regarding the copyright issues. Further comments will be given in the following sections. Therefore, the reviewer suggests the current version of manuscript is not suitable for publication in Water. However, the manuscript is still a good fit for a book chapter.

Response 2: As the reviewer notes, this is a review article summarizing the evolution of subsurface drainage systems over the centuries. As such, the paper’s hypothesis is that a careful review and consideration of the past developments in drainage systems underscore their importance to agricultural production in many parts of the world, and moreover, that failure to adequately consider possible drainage issues when developing agricultural production may lead to its eventual demise and damage to the local ecosystem.  We have added relevant text in this regard.

With respect to current drainage problems and associated water quality impacts– we have added more contemporary research and discussion about drainage system design with respect to water quality.

The matter of copyright on many of the figures is well considered.  Most figures are taken from the USDA and other such government publications, which are free in the public domain and have no copyright restrictions.  The remainder from the web remain unknown.  In all cases, appropriate citation of the picture source is provided.

Point 3:    The logic flow in introduction needs some improvement. The current version does not provide enough materials and discussion for research question and hypothesis.

Response 3: We have revised various parts of the text to improve reading flow and as noted above, added a hypothesis statement that sets the purpose for the review article.

Point 4:    A critical concern is the right to reproduce materials from other publications. For example, the manuscript employed and quoted many materials, i.e. figures (Fig. 23 & 24) and original content (Ln 150-157 and many others). Have the authors requested to reproduce these materials from the copy-right holders? If so, please attached the paperwork to grant the rights for reproduction from the publishers.

Response 4: The matter of copyright on many of the Figures is well considered. Most Figures are taken from the USDA and other such government publications, which are free in the public domain and have no copyright restrictions. The remainder from the web remain unknown. In all cases, appropriate citation of the picture source is provided. It is noted also, that the Figures 6(f, g, h, i) and 23, 24 are published (are incorporated in the main text) by permission of Alterra-ILRI Publisher. Please, see the captions of the Figures 6, 23 and 24.

Point 5:    Although the manuscript provides detailed review of subsurface drainage, the current version does not discuss the existing demand and problems that can be addressed by the development of materials and installation methods. For example, how to select the suitable synthetic envelops (geotextiles) in specific regions with the help of advanced techniques? In addition, to help reduce the dissolved nutrient export from agricultural drainage systems, how does the installation equipment become compatible with on-site treatment practices (e.g. bioreactors or saturated buffers)? The absence of such discussion substantially reduces the value of this excellent review.

Response 5: The first sentence above is not clear to us. We have described modern subsurface drainage installation techniques and materials. It is beyond the scope of this review paper to advocate for particular envelope materials for specific soils/regions. Rather, we have described the different envelope materials, something about the nature of their origin and where they were applied.

With respect to drainage system design and installation vis a vi water quality impacts, we have added a section discussing the water quality impacts of drainage system designs/installation. Further discussion of drain-water treatment to remove contaminants is beyond the scope of the review.

However to address the related intent of the review’s comments, we have added a brief section near the end of the article about drainage design and associated drain-water quality.

Point 6:    One of the missing part for drainage system is the trafficability (Ln 43-44). In many areas, machinery to plant and harvest crops require drainage to provide suitable trafficability.

Response 6: The text is corrected as follows: “(b) maintenance of groundwater levels for adequate root zone aeration and trafficability;” (Clean version, Lines 45 & 46) instead of “(b) maintenance of groundwater levels for adequate root zone aeration;” (original version, Lines 45 & 46)

Point 7:    Ln 124-128, “lacking drainage pipes … to the rivers”. As the pipes did not exist in the ancient times, the reviewer suggests removing this part and directly jumping to the first discovery of pipes in Ln 128.

Response 7: The part of the main text included from Line 124 to Line 128 is corrected as follows:

“Due to the lack of the drainage pipes, the early subsurface drains probably were trenches partially filled with highly permeable materials, such as gravel and stones, or permeable, voluminous substances such as bundles of small trees and shrubs tied together at the bottom of the trench [8].” (Clean version, Lines 163-166).

instead of

“Lacking drainage pipes, early drains were trenches partially filled with highly permeable materials such as gravel and stones, or permeable, voluminous substances such as bundles of small trees and shrubs tied together at the bottom of the trench [6]. These drains were used most likely as ditches for channeling floodwaters back to the rivers.” (original version, Lines 124 - 128)

Point 8:    Ln 145-146, “Columella states both … narrow”. Surface drainage is relevant to your study in this manuscript.

Response 8: The authors cannot understand the message of the Reviewer. Nevertheless, they corrected this part of the text as follows:

“Columella stated covered drains are to be made sloping at the sides and the bottom to be made narrow.” (Clean version Lines 60-61)

instead of

“Columella states both open and covered drains are to be made sloping at the sides, and the bottom to be made narrow.” (original version Lines 145 & 146)

Point 9:    Ln 133-157, Please be careful that this section describes the materials of subsurface drain pipes. This paragraph is mostly off the topic.

Response 9: This part of the main text included from Line 133 to Line 157 is transferred in the introduction of the manuscript after Line 47 (original version) (please see clean vesion Line 48-72). At the end of this text, we add the following: “Recently Valipour et al. (2020) presented a review article that traces the evolution of the main foundings on agricultural drainage technologies through the centuries until the present. This historical review reveals valuable insights into ancient hydraulic technologies and the management of irrigation and drainage over the years”. (Clean version Lines73-76

Point 10:  Ln 158, please remove the irrelevant information regarding surface drainage.

Response 10: We state in the manuscript (original version Lines 158-160): Romans were using open drains to remove ponded surface water and closed drains for removing surplus water from the soil itself.” The authors cannot understand the message of the Reviewer. We consider that the completeness of the subject requires us to mention that the Romans were using not only closed drains but and open ones. The reviewer examines the matter within a very narrow framework, something that does not find us in agreement

Point 11:  Ln 164-166, repeated with Ln 128.

Response 11:

  1. The part of main text from Line 128 to Line 129 is corrected as follows: “Perhaps the oldest known drain pipes were discovered in the Lower Indus river valley, they were fabricated about 2000 BC [12], or are approximately 4,000 years old.” (Clean version Lines 166-168).

instead of

Lines 128 & 129: The oldest drain pipes discovered are approximately 4,000 years old and were located in the Lower Indus river valley [10]. (original version, Lines 128 & 129).

  1. We delete Lines 164-166. “The first drainage pipes were discovered in the Lower Indus River valley; they were fabricated about 2000 BC [10] and so, they are about 4000 years age old.”

Point 12:  Ln 183-189, the reviewer can not understand the message to be delivered from this paragraph.

Response 12: The part of the main text included from Line 183 to Line 189 is corrected as follows:

“During the 17th century, subsurface drainage system installation in the United Kingdom included trenches filled with bushes or stones that were further developed on a larger scale across the Netherlands [21].” (Clean version Lines 195-197).

instead of

“Elliot [21] stated that field and farm drainage by means of the small open-ditch method was largely supplemented and, in many cases, supplanted by the covered trench or underdrain systems. Trenches in which were placed stones or brush to serve as a water conduit and covered with earth were employed a hundred years before tiles were known, and demonstrated conclusively, in many instances, that underdrains were superior to open ditches. During the 17th century, subsurface drainage system installation in the United Kingdom included trenches filled with bushes or stones that were further developed on a larger scale across the Netherlands [22].” (original version, Lines 183-189)

Moreover, the main text included from Line 195 to Line 198 is modified as follows:

“Whilst the clay working industry is an ancient industry dating to unrecorded history, the first clay drain pipes were used on the estate of Sir James Graham at Northumberland in England, in 1810 and they appear to have been the standard tile for about thirty years [20, 24]. The invention of clay pipes was marked an important epoch in the history of drainage [23]. The earliest form of tiles introduced for drainage purposes was the horseshoe shaped tile, so called due from its shape. .” (Clean version, Lines 203-207).

instead of

“Whilst the clay working industry is an ancient industry dating to unrecorded history, the first clay drain pipes were used on the estate of Sir James Graham at Northumberland in England, in 1810 and they appear to have been the standard tile for about thirty years [20, 21]). The earliest form of tiles introduced for drainage purposes was the horseshoe shaped tile, so called due from its shape.” (Original version, Lines 195-198)

Point 13:  Ln 209, the timeline is very confusing here. As the heading mentioned this section reviewed materials from 1800-1940s, why this type of tiles dated back to 1760?

Response 13: We corrected the inconsistent heading of the subsection “2.3. Drainage pipes from 1800 until ‘40s” (original version, Line 194) in “2.3. Drainage pipes from 1700 until ‘40s” (Clean version Line 202).

Point 14:  Ln 223, “…several years later…” please specify the exact year.

Response 14: It is corrected as follows: “In 1903, Elliott [30]…” (Clean version Line 232).

 Point 15:  Ln 237, “… by some 70%”, please revise this.

Response 15: We carefully examined the international literature and found the followings.

Robinson (1986, p. 97) stated: "….in I845 Thomas Scragg invented a machine for extruding drainage tiles, which brought their price down by about 70 per cent".

Robinson repeated exactly the same proposal 4 years later (Robinson 1990 p.14).

Since then, many articles, books, sites, etc., repeated the aforementioned.

In 2009, Minaev and Maslov (2009, p. 107) stated: “Tomas Scragg buit an exruding machine in 1845, which produced round clay pipes quickly reducing the cost from £21 to £16 per thousand”. Consequently, according to the Minaev and Maslov (2009, p. 107) the cost reduction was only 23.81% {=100*(21-16)/21}.

Maslov (2009, p. 3) stated “the revolution in drainage construction dates back to 1845, when Thomas Seragg invented an extruding machine that produced round clay pipes quickly, reducing cost”. Beauchamp (1987, p. 15) stated: “This new machine greatly reduced the cost of drain tile and led to its increased use”. Consequently, there is a confusion concerning the reducing cost of the drainage tile that produced the Thomas Scragg machine. We consider it is difficult to answer which of these points of views is the correct.

That's why we decided to correct the text as follows:

“The revolution in drainage construction dates back to 1845, when Thomas Scragg invented an extruding machine that produced round clay pipes quickly, reducing significantly their cost of production [14, 33, 34, 35, 36]. This invention constituted an important factor in spreading the use of the drainage worldwide.” (Clean version Lines 245-248.

instead of

“The revolution in drainage construction dates back to 1845, when Thomas Scragg invented an extruding machine that produced round clay pipes quickly, reducing significantly the cost by some 70% [33, p. 14]. This invention constituted a revolution in drainage construction and as consequence; it led to an increase in its use [12]” (original version, Lines 236-239)

Point 16:  Ln 240, “For the next county, … in all counties” The reviewer suggests being really cautious about this statement. Not all the counties use the same material for subsurface drainage.

Response 16: We correct the text as follows:

“For the next century, ceramic (tile) pipes became the basic means of drainage in several countries, such as Jordan, Spain, Pakistan, and a number of other countries”. (Clean version Line 249 - 251)

instead of

"For the next century, ceramic (tile) pipes became the basic means of drainage in all countries. (original version, Lines 240 & 241).

Point 17:  Ln 242-244, the reviewer can not understand the message from this paragraph.

Response 17: The part of the main text included from Line 242 to Line 244 is corrected as follows:

“In 1830, for the first time, Portland cement was used to make drain tile by hand [1]. The manufacturing of concrete drain pipes started in the USA the 1862, when David Ogden developed a suitable machine for making drain pipes from cement and sand [19]. This machine could make pipe with an inside diameter of 5.715 cm (2-1/4 in) to 60.96 cm (24 in). Obviously, the use of concrete drain tiles dominated in areas where good clay was not available, while in the other areas, economic factors determined the choice of one of these materials.” (Clean version Lines 252 - 257).

instead of

“In 1830, for the first time, Portland cement was used to make drain tile by hand [1]. However, significant boost in agricultural drainage was the manufacturing of the first drainpipe from sand and cement in USA, in 1862 [17].” (original version, Lines 242-244)

Point 18:  Ln 245, “… both clay and concrete pipes were the main drainage materials…” this statement conflicts with the statement in Ln 240. Please clarify this confusion.

Response 18: It corrected. Please, see Point 16 and Response 16.

 Point 19:  Ln 263-268, it seems that this paragraph does not add too much information in addition to the sentence in Ln 261-262, “Since the 1980s, …for drainage pipes”.

Response 19: We corrected and modified the text included between the Lines 261-268 as follows:

“Nowadays, the corrugated plastic drains are made of PVC, high-density polyethylene (HDPE) and polypropylene (PP). Since the 1980’s, corrugated high-density polyethylene (HDPE) and corrugated polyvinyl chloride (PVC) pipes are the preferred standards for drainage pipes [13, 39]. The choice of one of these materials is determined by economic factors. It is worth noting while PVC and HDPE, as raw materials, have different physical properties, it is possible to make durable pipes of good quality from either materials. In the United Kingdom the drains are made of HDPE and a minority of PP, while in the rest Europe corrugated drains are mainly made of PVC. In the USA and Canada most drainpipes are made of HDPE, mainly because of the low price of the raw material [8].” (Clean version Lines 276 - 283).

instead of

 Since the 1980’s, corrugated high-density polyethylene (HDPE) and corrugated polyvinyl chloride (PVC) pipes are the preferred standards for drainage pipes [11, 36].

Nowadays, the corrugated plastic drains are made of PVC, high-density polyethylene (HDPE) and polypropylene (PP). Economic factors determine the choice of one of these materials. While PVC and HDPE, as raw materials, have different physical properties, it is possible to make durable pipes of good quality from either materials. Stuyt et al. [6] noted that "in Europe, corrugated drains are mainly made of PVC except for the United Kingdom, where they are made of HDPE and a minority of PP. In the United States and Canada, most drainpipes are made of HDPE, largely because of the low price of the raw material". (original version, Lines 261-268).

Point 20:  Ln 309-349, these intensive definitions of envelop seems off the topic of your study.

Response 20:

After the comments of the Reviewer 1, Point 20 and the Reviewer 2, Point 5 we corrected and modified the text included between the Lines 309-349 as follows (Clean version Line 324 – Table 1-Line 330):

“During the past several decades, several definitions have been used to describe drain envelopes that include its functions relative to soil and trench hydraulics, trench stability and as bedding materials for drainpipes. Table 1 shows various definitions which presented in the international literature and the corresponding citation.

Table 1. Various definitions, which presented in the international literature and the corresponding citation

Citation

Definition

USDA-SCS [48]

Permeable materials placed around the drains for the purposes of improving flow conditions in the area immediately surrounding the drain, and to improve pipe bedding conditions.

Ritzema [49]

Material placed around pipe drains to serve one or a combination of the following functions: (i) to prevent the movement of soil particles into the drain; (ii) to lower entrance resistances in the immediate vicinity of the drain openings by providing material that is more permeable than the surrounding soil; (iii) to provide suitable bedding for the drain; (iv) to stabilize the soil material on which the drain is being laid.

Cavelaars et al.  [46]

Material placed around pipe drains to perform one or more of the following functions, namely (i) filter function, i.e. to prevent or restrict soil particles from entering the pipe where they may settle and eventually clog the pipe; (ii) hydraulic function, i.e. to constitute a medium of good permeability around the pipe and thus reduce entrance resistance; and (iii) bedding function, i.e. to provide all-round support to the pipe in order to prevent damage due to the soil load.

Stuyt and Willardson [50]

Porous material placed around a subsurface drain to protect the drain from sedimentation and improve its hydraulic performance.

Vlotman et al. [39]

Porous material placed around a perforated pipe drain to perform one or more of the following functions: (i) filter function namely to provide mechanical support or restraint of the soil, at the drain interface with the soil, to prevent or limit the movement of soil particles into the drainpipe where they may settle and eventually clog the pipe; (ii) hydraulic function namely to provide a porous medium of relatively high permeability around the pipe to reduce entrance resistance at or near the drain openings; (iii) mechanical function namely to provide passive mechanical support to the pipe in order to prevent excess deflection and damage to the pipe due to soil load; and (iv) bedding function namely to provide a stable base to support the pipe in order to prevent vertical displacement due to soil load during and after construction. The functions (iii) and (iv) can only be achieved with gravel and sand envelopes.

Wright and Sands [44]

Material placed around a drainpipe to provide either hydraulic function, which facilitates flow into the drain, or barrier function, which prevents certain sized soil particles from entering the drain.

USDA-NRCS [51]

Generic term that includes any type of material placed on or around a subsurface drain for one or more of the following reasons: (i) to stabilize the soil structure of the surrounding soil material, more specifically a filter envelope; (ii) to improve flow conditions in the immediate vicinity of the drain, more specifically a hydraulic envelope; (iii) to provide structural bedding for the drain, also referred to as bedding.

Nijland et al.  [13]

Porous material placed around a perforated drain pipe: (a) to prevent or restrict soil particles from entering the drain pipe where they may settle and eventually clog the pipe (filter function); (b) to provide a porous medium of relatively high permeability around the pipe to reduce entrance resistance (hydraulic function).

Stuyt et al. [8]

Porous material placed around a subsurface drain, to protect the drain from sedimentation and improve its hydraulic performance.

Hereafter, we adopt the aforementioned definition of Vlotman et al. [39]

Point 21:  Ln 341-359, another example of intensive quotes from existing literature. The reviewer defers this issue to the editor.

Response 21: It is corrected. Please see Point 20 and Response 20.

Point 22:  Ln 417-424, further in-depth review of recent development of synthetic envelop is needed as this is the recent hot-spot in related research.

Response 22:

Point 20 is relative with Point 5.

As we have noted in our Response #5, we have described modern subsurface drainage installation techniques and materials. We agree with the Reviewer that development of synthetic envelop is the recent hotspot in the related research. However, this topic is beyond the scope of this review paper and consequently, it does not advocate in-depth review of it.

Point 23:  Ln 607, it might be better to replace “epilogue” with “conclusion” (or similar).

Response 23: Done. We replaced the inconsistent heading “Epilogue” with “Conclusion”.

Point 24:    After reading this manuscript for multiple times, the reviewer still thinks the current version is logic, well-organized and carefully-prepared. However, due to the lack of key research question and problems and demand from recent studies, the reviewer suggests substantial revision is needed before the publication.

Response 24:  Please see our previous comments and revisions that we hope address this last concern by the reviewer overall. Again, we note that this is a REVIEW article rather than a research investigation and the hypothesis is broad in scope as noted above in Response #2.

 All changes are tracked in the revised version and a “clean” copy is also provided. On behalf of the co-authors

Yours sincerely,

Dr. Stavros Yannopoulos

Emeritus Professor

Reviewer 2 Report

This manuscript provides an interesting and informative historical review of field drainage, including the evolution of the technology and materials used. I am recommending it for publication with very minor modifications.

Overall an excellent manuscript and worthy of publication

I have two conceptual issues. Firstly no reference to the potential of salty and nutrient rich water (and possibly pesticide rich water) to pollute water ways - this is a major problem in p[arts of the world.

Secondly, there is no reference to bio drainage - the use of plants (eg plantations) to soak up surplus water. this has been developed in parts of Australia and elsewhere for salinity remediation (eg for Perths water supply catchments) and in parts of the wheat belt of WA.

Most of these could be found by a good editor and proof reader, but here are a few:

Page 5 line 196 and line209 dates used appear inconsistent - this needs correction or explanation

Section 3.1 is a bit repetitive and could be consolidated.

Page 12 line 475 'trenches' is spelt incorrectly

Page 13 line 508 missing word- assume 'on' and line 517 use of this and the in same role in sentence - delete one

Page 15 559 'they can (a) to' suggest delete to

Page 17 611-626 important to note that much of the technological improvement was borrowed and modified from wider technological programns eg improvement in machinery and materials

one small omission is the absence of reference to bio-drainage - the sue of plants

see for example below

[BOOK] Biodrainage: principles, experiences and applications

AF Heuperman, AS Kapoor, HW Denecke - 2002 - books.google.com Biodrainage relies on vegetation, rather than mechanical means, to remove excess water. It
is economically attractive because it require only an initial investment for planting the
vegetation, and when established, the system could produce economic returns by means of … Cited by 127 Related articles

Author Response

Response to Reviewer 2 Comments

Manuscript ID: water-792752

Title: Evolution of the materials and methods used for subsurface drainage of agricultural lands from antiquity to the present

Authors:  Stavros Yannopoulos*, Mark E. Grismer, Khaled Bali, Andreas Angelakis

Journal: Water

Dear Reviewer,

we express our gratitude for the excellent revision work you made. We have modified the manuscript in accordance with your comments. The answers to your comments are in what follows.

Point 1:    This manuscript provides an interesting and informative historical review of field drainage, including the evolution of the technology and materials used. I am recommending it for publication with very minor modifications. Overall an excellent manuscript and worthy of publication

Response 1: The authors would like to thank the anonymous reviewer for his time, efforts and constructive comments that helped us to re-consider and to improve our manuscript.

Point 2:    I have two conceptual issues. Firstly no reference to the potential of salty and nutrient rich water (and possibly pesticide rich water) to pollute water ways - this is a major problem in p[arts of the world.

Response 2: We added in the main text a new section entitled "5. Drainage System Design/Installation Effects on Drain-Water Quality" (Clean version Lines 590-639)

The risk posed by discharge of drain water containing nutrients, salinity and other contaminants to the ecology of receiving waters, such as streams and wetlands, is of growing concern and must now be addressed by those responsible for management of the agricultural production, whether irrigated or otherwise. For example in parts of the San Joaquin Valley of California, farmers dispose of low-quality drain water by evaporation from constructed collection ponds, or re-apply the saline drainage water to successively more salt-tolerant crops until the remaining collected brine is used in solar energy generation ponds. The drain-water evaporation ponds have a limited future associated with the fate of the accumulating salts precipitating as the water evaporates (Tanji and Grismer 1987) [87]. On the other hand, in more humid regions, agricultural drain water can safely pass through wetlands with minimal adverse impacts while simultaneously improving the chemical and biological quality of the water. However, more work is needed to define acceptable drain-water quality criteria for wetland and stream disposal.

Subsurface drainage systems collect the root zone drainage water that presumably reflects the water quality (salinity, mineral dissolution, excess applied nutrients, herbicides and pesticides) of the root zone leachate and depends in part on the relative efficacy of the drainage system in collecting this leachate across the soil profile width between drain laterals. Drainline spacing and depth together with varying soil permeability control the water flow paths within and below the root zone that then influences the water quality of the collected drainage water [88]. Drain-water salt load-flow relationships are a measure of the drainage system capacity and efficacy to extract salt and reduce the salinity of the root zone while also indicating something of the possible nutrient and other pollutant capture by the drainage system. Theoretically, when the salt concentration is constant, there is a direct, usually linear relationship between salt load and drain discharge or flow rate; that is, as the flow rate increases, there is an increase in the salt load, partly due to the increase in flow.  Grismer et al. [7] noted that the previous investigators Frank E. Robinson and James N. Luthin found that when originally installed in the 1960s, the bituminous fiber and clay tile drainage systems yielded a decreasing salt load with increasing flow as a result of “a unique combination of hydraulic conductivities along each tile line” such that “areas of high hydraulic conductivity tended to dilute the salinity in the areas of low hydraulic conductivity”. However, later Grismer et al. [7] found a linearly increasing relationship between salt load and drain flows as expected though a newly installed drainage system yielded a roughly 20% smaller load for the same drain flowrate as compared to that installed some 20 years prior. Grismer [88] and Guitjens et al. [89] underscored that “as mechanisms of in-situ water mixing, salt mobilization, and solute transport are better understood, drainage designs should include criteria that make it possible for management to manipulate the quantity and quality of drainage discharge”.

Ideally, mechanistic models should give reasonable estimates of the mass of water, salt, and the individual salt components involved in subsurface water movement. When water flows downward through the vadose zone to the water table and becomes ground water, the saturated conditions may change the chemical processes. Even process-oriented models of the vadose and saturated ground-water zones are necessarily based on simplifying assumptions, such as chemical equilibrium, which may not occur in reality. Accurate data on vadose zone and aquifer minerals, pH and redox status, and the heterogeneous nature of the media are difficult to acquire. Grismer [88] used steady state and transient numerical simulations to model the effects of depth and spacing of drains on drain discharge and quality in terms of salt concentration and load. Under conditions of increasing salinity with soil depth, results of the steady state model yielded increased drain water salinity as either drain spacing or depth increased, but the increase with drain depth was more pronounced. Flow paths and groundwater displacement remained somewhat invariant in transient flow. The deeper drains and wider spacing had greater flow, salt concentration, and salt load. Regardless, from a water management perspective, decreasing drainage flows through improved irrigation application efficiency will likely reduce contaminant loads to receiving waters accepting drainage discharge, whether surface or groundwater.

Point 3:    Secondly, there is no reference to bio drainage - the use of plants (eg plantations) to soak up surplus water. this has been developed in parts of Australia and elsewhere for salinity remediation (eg for Perths water supply catchments) and in parts of the wheat belt of WA.

Response 3: We added in the main text a new section entitled 6. Alternative drainage technique to control waterlogging and salinity – Biodrainage  (Clean version Lines 640-692)

The conventional drainage technologies (surface drainage and subsurface drainage) have two basic drawbacks, namely, they are costly and they generate drainage effluents, which will have to be either carefully re-used or safely disposed of. It is worth noting the management of drainage effluents has become an important issue around the world. Moreover, the lack of finance and the inadequate maintenance often impedes their installation and sustainability, respectively. The aforementioned reasons prompted the search for alternative approaches to maintaining sustainable agriculture in the long run. These alternatives must be effective, affordable, socially acceptable and environmentally friendly and not cause degradation of natural land and water resources.

The technology known as biodrainage is a quite new promising technology which is environmentally friendly and of less cost.

The term biodrainage first time documented by Gafni [90], however earlier, Heuperman [91] used term bio-pumping to describe the use of plants for water table control. It has also included the term of biodisposal, which refers to the use of plants for final disposal of excess drainaged waters. Biodrainage could be used in irrigation schemes to control the rising water table and avoid waterlogging and salinization [92].

Biodrainage processes can be classified based on land use context as: (a) Dryland/rainfed systems: (i) recharge control, (ii) groundwater flow interception and (iii) discharge enhancement; and (b) Irrigated systems: (i) water table control, (ii) channel seepage interception, and biodrainage cum conventional drainage systems [92].

Plants and especially trees with high evapotranspiration rates should be used. The evapotranspiration principle of plants is used in biodrainage treatment to reclaim such problematic areas sustainably (Figure 25). Evergreen broad-leaved species recorded high evapotranspiration rate could be used for reclamation of waterlogged soils. Sarvade et al. [93] reported that short rotation fast growing tree species, such as Salix spp., Eucalypt spp., Acacia spp., Albizia spp., Terminalia spp., Prosopis spp., and Populus spp. were the suitable species for biodrainge technology. By using such trees, the elevated ground water table of an area could be reduced [92]. Agri-silviculture, agri-horti-silviculture, silvi-pasture, multipurpose woodlots, strip plantation and boundary plantations were widely used for reclamation of saline-waterlogged conditions of India. In agri-silviculture system, Eucalypt spp. based agro-forestry systems are widely used for reclamation of waterlogged areas as compared to other woody plant based systems.

Figure 25. Concept of biodrainage (modified from Ram et al. [94]).

In India highly evapotranspirating tree species selected to mitigate waterlogging conditions [95]). In a study in Rajasthan in northern India an average annual rate of evapotranspiration was estimated as 3446  mm (which is about 1.4 Class A pan) from a 25-ha land planed with Eucalyptus camaldulensis, Acacia nilotica, Prosopis cineraria and Ziziphus spp. in a density of 1900 trees/ha [96]. Also in another study where numerous plant systems were tested, an Eucalypt based agri-silviculture system gave good resuls with 0.84–0.86 m total drawdown of ground water table in 3 years period and is widely used vegetation system in India [93].

One special application of the biodrainage technology is the amelioration of waterlogged soils during the initial reclamation or ripening phase of ‘new’ land development. Vegetation with a vigorous, deep and extensive root system is used to dry out waterlogged soil profiles. For example, in the Netherlands land to be reclaimed from the sea is sown with reed while a few centimeters of water remain [92]. Such seeding accelerates the ripening process.

However, experimental results showed that the biodrainage was highest when the ground water salinity was lowest. Its magnitude decreased with increase in salinity of the ground water (Ram et al. 2008) [94]. Ιn Australia and elsewhere it is widely accepted that in discharge situations, enhanced evapotranspiration biodrainage sites will eventually succumb to salinity, unless some form of conventional drainage is installed to control salt balance to the vegetation’s root zone by removal of saline drainage effluent [97]. In that case plants of salt tolerance (e.g. Acacia kempeana and Eucalyptus tereticornis) should be used [98] and or for the final salt balance control disposal system is required  [92].

Point 4:    Page 5 line 196 and line209 dates used appear inconsistent - this needs correction or explanation

Response 4: We corrected the inconsistent heading of the subsection “2.3. Drainage pipes from 1800 until ‘40s” (original version, Line 194) in “2.3. Drainage pipes from 1700 until ‘40s” (Clean version, Line 202).

Point 5:       Section 3.1 is a bit repetitive and could be consolidated.

Response 5: After the comments of the Reviewer 1, Point 20 and the Reviewer 2, Point 5 we corrected and modified the text included between the Lines 309-349 as follows (Clean version Lines 324-330)

During the past several decades, several definitions have been used to describe drain envelopes that include its functions relative to soil and trench hydraulics, trench stability and as bedding materials for drainpipes. Table 1 shows various definitions which presented in the international literature and the corresponding citation.

Table 1. Various definitions, which presented in the international literature and the corresponding citation

Citation

Definition

USDA-SCS [48]

Permeable materials placed around the drains for the purposes of improving flow conditions in the area immediately surrounding the drain, and to improve pipe bedding conditions.

Ritzema [49]

Material placed around pipe drains to serve one or a combination of the following functions: (i) to prevent the movement of soil particles into the drain; (ii) to lower entrance resistances in the immediate vicinity of the drain openings by providing material that is more permeable than the surrounding soil; (iii) to provide suitable bedding for the drain; (iv) to stabilize the soil material on which the drain is being laid.

Cavelaars et al.  [46]

Material placed around pipe drains to perform one or more of the following functions, namely (i) filter function, i.e. to prevent or restrict soil particles from entering the pipe where they may settle and eventually clog the pipe; (ii) hydraulic function, i.e. to constitute a medium of good permeability around the pipe and thus reduce entrance resistance; and (iii) bedding function, i.e. to provide all-round support to the pipe in order to prevent damage due to the soil load.

Stuyt and Willardson [50]

Porous material placed around a subsurface drain to protect the drain from sedimentation and improve its hydraulic performance.

Vlotman et al. [39]

Porous material placed around a perforated pipe drain to perform one or more of the following functions: (i) filter function namely to provide mechanical support or restraint of the soil, at the drain interface with the soil, to prevent or limit the movement of soil particles into the drainpipe where they may settle and eventually clog the pipe; (ii) hydraulic function namely to provide a porous medium of relatively high permeability around the pipe to reduce entrance resistance at or near the drain openings; (iii) mechanical function namely to provide passive mechanical support to the pipe in order to prevent excess deflection and damage to the pipe due to soil load; and (iv) bedding function namely to provide a stable base to support the pipe in order to prevent vertical displacement due to soil load during and after construction. The functions (iii) and (iv) can only be achieved with gravel and sand envelopes.

Wright and Sands [44]

Material placed around a drainpipe to provide either hydraulic function, which facilitates flow into the drain, or barrier function, which prevents certain sized soil particles from entering the drain.

USDA-NRCS [51]

Generic term that includes any type of material placed on or around a subsurface drain for one or more of the following reasons: (i) to stabilize the soil structure of the surrounding soil material, more specifically a filter envelope; (ii) to improve flow conditions in the immediate vicinity of the drain, more specifically a hydraulic envelope; (iii) to provide structural bedding for the drain, also referred to as bedding.

Nijland et al.  [13]

Porous material placed around a perforated drain pipe: (a) to prevent or restrict soil particles from entering the drain pipe where they may settle and eventually clog the pipe (filter function); (b) to provide a porous medium of relatively high permeability around the pipe to reduce entrance resistance (hydraulic function).

Stuyt et al. [8]

Porous material placed around a subsurface drain, to protect the drain from sedimentation and improve its hydraulic performance.

Hereafter, we adopt the aforementioned definition of Vlotman et al. [39]

Point 6: Page 12 line 475 'trenches' is spelt incorrectly

Response 6: In the revised version, “trneches” has be written correctly “trenches” (Clean version, Line 456).

Point 7:       Page 13 line 508 missing word- assume 'on' and line 517 use of this and the in same role in sentence - delete one

Response 7: In the page 13 Line 508 of the original version, the missing word is “on”. In the revised version of the manuscript the erroneous text: “Commenting the development of excavating machinery” has be written correctly: “Commenting on the development of excavating machinery” (Clean version, Line 489).

Also, in the Page 13 Line 517 of the original version the erroneous text: “…characterized the this advance as…” has be written correctly: “…characterized this advance as…” (Clean version, Line 498).

 Point 8:       Page 15 559 'they can (a) to' suggest delete to

Response 8: After the comment of the Reviewer, the main text included from the Lines 558 to Line 561 of the original version: “Trenchers are manufactured in various types and sizes with a large range of abilities, namely they can: (a) to install pipes to a depths of about 3 m, in hard or stony soil, in unstable subsoils and/or under the groundwater level; (b) to make trenches up to 0.50-0.60 m wide, and (c) work in soils with hard layers [11].”

corrected as

 “Trenchers are manufactured in various types and sizes with a large range of abilities, in particular: (a) to install pipes to a depths of about 3 m, in hard or stony soil, in unstable subsoils and/or under the groundwater level; (b) to make trenches up to 0.50-0.60 m wide, and (c) to work in soils with hard layers [13].” (Clean version, Lines 539-542).

Point 9:       Page 17 611-626 important to note that much of the technological improvement was borrowed and modified from wider technological programns eg improvement in machinery and materials

Response 9:

In order to take into account the aforementioned comment of the Reviewer the part of the main text included from Line 611 to Line 626 corrected as follow (Clean vesion Lines 697-718):

Over the last two centuries there has been considerable progress made by engineers and scientists in the field of drainage. It is worth noting, since 1950s, the new types of installation equipment, materials and techniques were revolutionary for the drainage industry and contributed to the improvements of the drainage systems installation quality and durability. Subsurface drainage methods, materials, technologies, and installation were modernized much more through innovative research and development between 1960 and 1975 than during the previous centuries. In particular, since 1960s and thereafter, the drainage technology has changed rapidly with enormous improvements in drainage tubing, machinery and methods of installation, drainage envelopes (natural and manufactured), and techniques for quality assurance and control. Additive factors that have influenced and contributed to the progress and spread of land drainage are (a) the improvement of land drainage theory; (b) the invention of a new generation of computers with many possibilities in memory and speed; (c) the development of computer simulation models which use the drainage theory to describe the performance of drainage and related water management systems; (d) the development of new tools such as Geographical Information Systems (GIS) and Computer Aided Design (CADD) to the systematic planning and design of drainage systems. It should be noted that this improvement resulted due to the better comprehension of the fundamental processes and relationships through a long time of research and practice, as well of the improved methods that were applied to the effects on the crop: (a) of the soil salinity and (b) of excessive and deficit soil water conditions. The aforementioned events in combination with the development and the transformation of agricultural drainage contributed to the expansion of agricultural drainage worldwide. Nowadays, the agricultural drainage is considered as an integral part of the total water management and not just as surplus excess water and salt removal, as in the past. As Skaggs and van Schilfgaarde [85] pointed out “the systematic planning and design of drainage systems, particularly large-scale projects, has been impacted by the development of new tools such as Geographic Information Systems (GIS) and Computer Aided Design (CADD)”.  

NOTE: Citation No 85 deleted.

Point 10:  one small omission is the absence of reference to bio-drainage - the sue of plants see for example below. [BOOK] Biodrainage: principles, experiences and applications AF Heuperman, AS Kapoor, HW Denecke - 2002 - books.google.com Biodrainage relies on vegetation, rather than mechanical means, to remove excess water. It is economically attractive because it requires only an initial investment for planting the vegetation, and when established, the system could produce economic returns by means of… Cited by 127 Related articles.

Response 10:

It is corrected. Please, see Point 3 and the Response 3. All changes are tracked in the revised version and a “clean” copy is also provided. On behalf of the co-authors

Yours sincerely,

Dr. Stavros Yannopoulos

Emeritus Professor

Reviewer 3 Report

Very interesting research and manuscript. Very useful to improve a wide view about drainage. Congratulations!

My unique suggestions (although I am not the English expert).

1) Line 40, you use "and/or"; I think that it is not a very formal  the use of symbol "/"; I suggest to replace by "and, or, ", or, "and or"

2) The use of "etc." is used several times; I think that it is not very elegant in scientific writing; maybe replace by "for example", "among others"

Author Response

Response to Reviewer 3 Comments

Manuscript ID: water-792752

Title: Evolution of the materials and methods used for subsurface drainage of agricultural lands from antiquity to the present

Authors:  Stavros Yannopoulos*, Mark E. Grismer, Khaled Bali, Andreas Angelakis

Journal: Water

Dear Reviewer,

we express our gratitude for the excellent revision work you made. We have modified the manuscript in accordance with your comments. The answers to your comments are in what follows.

Point 1:    Very interesting research and manuscript. Very useful to improve a wide view about drainage. Congratulations! My unique suggestions (although I am not the English expert).

Response 1: The authors would like to thank the Reviewer for his time, efforts and constructive comments that helped us to re-consider and to improve our manuscript.

Point 2:    Line 40, you use "and/or"; I think that it is not a very formal the use of symbol "/"; I suggest to replace by "and, or, ", or, "and or"

Response 2: In fact, the grammatical conjunction “and/or” appears in the manuscript two times (Lines 40 and 560). According to the Merriam-Webster Dictionary (https://www.merriam-webster.com/dictionary/and%2For, assessed on May 13, 2020), the first known use of “and/or” is in 1853 and is used as a function word to indicate that two words or expressions are to be taken together or individually. Moreover, according to the Cambridge Dictionary (https://dictionary.cambridge.org/dictionary/english/and-or, assessed on May 13, 2020), “and/or” is a conjunction used to refer to both things or either one of the two mentioned either "and" or "or". There is nothing wrong with writing in our manuscript (Lines 38-40) “Nowadays, in several regions of the world some agricultural lands (e.g. coastal plains, river valleys, and inland plains) have limited natural drainage capacity due to poorly drained root zone soils and/or shallow water tables.”; and (Lines 559-560) “to install pipes to a depths of about 3 m, in hard or stony soil, in unstable subsoils and/or under the groundwater level”. The authors to the extent that they can know, they did not fall into their perception any criticize on "and/or", namely as ugly in style and ambiguous, while they have read this grammatical conjunction many times in various scientific articles, in their long time career.

Nevertheless, the authors accept the proposal of the Reviewer and they make the following changes in the main text:

1. (Original version, Lines 38-40): “Nowadays, in several regions of the world some agricultural lands (e.g. coastal plains, river valleys, and inland plains) have limited natural drainage capacity due to poorly drained root zone soils and/or shallow water tables.”

corrected as

 “Nowadays, in several regions of the world some agricultural lands (e.g. coastal plains, river valleys, and inland plains) have limited natural drainage capacity due to poorly drained root zone soils and or shallow water tables.” (Clean version, Lines 38 & 39).

2. (Original version, Lines 559-560): “to install pipes to a depths of about 3 m, in hard or stony soil, in unstable subsoils and/or under the groundwater level”

corrected as

“to install pipes to a depths of about 3 m, in hard or stony soil, in unstable subsoils and or under the groundwater level” (Clean version, Lines 540 & 541)

Point 3:    The use of "etc." is used several times; I think that it is not very elegant in scientific writing; maybe replace by "for example", "among others"

Response 3:

The adverb “etc.” is the abbreviation of Latin word “et cetera”, where “et“ means “and”, while “cetera” means “the rest”. According to the Cambridge Dictionary (https://dictionary.cambridge.org/dictionary/english/etc, assessed on May 13, 2020), it means "and other similar things". It is used to avoid giving a complete list, but after a list.

According to the grammatical rules: (a) it can be used only when unmentioned items are of the same type as the items mentioned earlier; (b) it indicates that there are other items in the list besides the ones we explicitly mention; (c) the abbreviation is more common than the full phrase in business and technical writing.

In fact, the abbreviation “etc.” appears in our manuscript ten times (Lines 67, 220, 441, 452, 494, 529, 530, 531, 548, 839) in a text of 11,532 words. The authors have read carefully the Instructions for Authors of the WATER Journal and they do not find limitations or instructions about the use of adverb “etc.”

In Line 220, the adverb "etc." is included in a quote from existing literature (Chamberlain [29]). In line 839 the use of "etc." concerns the citation number [82] (Curwen et al. 1797), due to the great number of co-authors (more than 10) in accordance with the Instructions for Authors of the WATER Journal.

Based on the above, with any respect, the authors find it difficult to understand the reviewer's claim:  "it is not very elegant in scientific writing".

Nevertheless, the authors tried to diminish the use of the adverb "etc." in the main text.

In particular:

1. (Original version, Line 67) “….America, Russia, etc.) to control…”

corrected as

“….America, Russia) to control…” (Clean version, Line 96)

2. (Original version, Line 441) “…hatchet pick, etc. Figure 6 shows…”

corrected as

“…hatchet pick, among others. Figure 6 shows…” (Clean version, Line 422)

All changes are tracked in the revised version and a “clean” copy is provided. On behalf of the co-authors

Yours sincerely,

 Dr. Stavros Yannopoulos

Emeritus Professor

Round 2

Reviewer 2 Report

The additions and revisions are great and satisfy my previous concerns - excellent manuscript

Author Response

Response to Academic Editor Notes

Manuscript ID: water-792752

Title: Evolution of the materials and methods used for subsurface drainage of agricultural lands from antiquity to the present

Authors:  Stavros Yannopoulos*, Mark E. Grismer, Khaled Bali, Andreas Angelakis

Journal: Water

Dear Editor,

we express our gratitude for your time, efforts and constructive notes that helped us to re-consider and to improve our manuscript. The answers to your notes are in what follows.

Point 1:    “42 salinity accumulation --> salt accumulation”

Response 1: Corrected. Please, see Line 42 of clean version.

Point 2:    “47 upward flowing --> upward flow”

Response 2: Corrected. Please, see Line 47 of clean version.

Point 3:    “57 mention draining --> mentioned drainage”

Response 3: Corrected. Please, see Line 57 of clean version.

Point 4:    “57 give --> gave”

Response 4: Corrected. Please, see Line 57 of clean version.

Point 5:    “58 directs --> directed”

Response 5: Corrected. Please, see Line 60 of clean version.

Point 6:    “61 "Also," is not formal wording in scientific literature. Please consider some of "Also," to other similar words such as "In addition," or "Moreover, "

Response 6: Corrected. Please, see Lines 62 and 64 of clean version.

Point 7:    “167 they --> which”

Response 7: Corrected. Please, see Line 168 of clean version.

Point 8:    “172 How does "closed drains" different from "subsurface drainage?"

Response 8: In the past, water enters into the tiles (closed drains, i.e. clay, cement pipes, etc.) through the joints or between the ends of the tiles [30, p. 38]. In the modern times, the pipes (closed drains) have to be permeable (perforated) or require openings at the joints of pipe sections so that water can enter them. The perforations or openings of the pipes should be: (a) As large as possible to limit the entry resistance of the water; (b) As small as possible to prevent the soil particles surrounding the pipes entering the pipe as a result of mobilisation by the water flow. If soil particles enter the pipe they will sediment in the pipe and obstruct the flow [39, p.1; 13, p. 70].

Point 9:    “195 subsurface drainage system installation --> installation of subsurface drainage system”

Response 9: Corrected. Please, see Line 196 of clean version.

Point 10:  “351 has considered --> considered”

Response 10: Corrected. Please, see Line 352 of clean version.

Point 11:  “421 such as are --> such as”

Response 11: Corrected. Please, see Line 422 of clean version.

Point 12:  “Figure 6 ia, ib, ic --> “

Response 12: The caption of Figure 6 corrected as follows:

“Figure 6. Digging and installation tools used in constructing tile drains by hand: (a) Ordinary Shovel  [61]; (b) Bottoming Spade [29]; (c) Bottoming Scoop [29]; (d) Ordinary Spade [61]; (e) Laying Hook [62]; (f) Correction Hook [13]*; (g) Drain Scoop [13]*; (h) Hoe [13]*; (i) Pipe tongs [13]*; (ia) Soil pincer [13]*; (ib) Hatchet Pick [25]; (ic) Hand Pick [63] (Conception, synthesis and processing have been made by Stavros Yannopoulos) (* by permission of Alterra-ILRI Publisher).

instead of

“Figure 6. Digging and installation tools used in constructing tile drains by hand: (a) Ordinary Shovel  [61]; (b) Bottoming Spade [29]; (c) Bottoming Scoop [29]; (d) Ordinary Spade [61]; (e) Laying Hook [62]; (f) Correction Hook [13]*; (g) Drain Scoop [13]*; (h) Pipe tongs [13]*; (i) Soil pincer [13]*; (j) Hatchet Pick [25]; (k) Hand Pick [63] (Conception, synthesis and processing have been made by Stavros Yannopoulos) (* by permission of Alterra-ILRI Publisher).”

Point 13:  “434 Road Grader, why capital?”

Response 13: Corrected. Please, see Line 435 of clean version.

Point 14:  “525 fuel --> combustion”

Response 14: Corrected. Please, see Line 526 of clean version.

Point 15:  “548 Figure 19 is not Fowler's Drain Plow.”

Response 15: Many thanks for this note. Sorry, it is our fault. Corrected. Please, see Line 550 of clean version. In fact, Figure 16 is Fowler's Drain Plow NOT Figure 19.

Point 16: “591 salinity --> salts”

Response 16: Corrected. Please, see Line 593 of clean version.

Point 17:  594 dispose of --> dispose

Response 17: Corrected. Please, see Line 596 of clean version.

Point 18:  “609 salt --> salts”

Response 18: Corrected. Please, see Line 610 of clean version.

Point 19:  “611 capture --> captured”

Response 19: Corrected. Please, see Line 613 of clean version.

Point 20:  “614 Frank E. Robinson and James N. Luthin --> , Frank E. Robinson and James N. Luthin, “

Response 20: Corrected. Please, see Line 616 of clean version.

Point 21:  “649 "promising" should be removed considering the restriction of application to non-saline land.”

Response 21: Corrected. Please, see Line 651 of clean version.

Point 22:  “660 Plants and especially trees --> Plants, especially trees, ‘

Response 22: Corrected. Please, see Line 662 of clean version.

Point 23:  “674 India an --> India, an”

Response 23: Corrected. Please, see Line 676 of clean version.

Point 24:  “698 noting, --> noting that, “

Response 24: Corrected. Please, see Line 700 of clean version.

Point 25:  ‘735 growth --> development of “

Response 25: Corrected. Please, see Line 737 of clean version.

Moreover, the Authors make the following corrections

Point (A)

We corrected Line 172 of clean version

“2.2. Drainage pipes from antiquity until about 1700”

instead of

“2.2. Drainage pipes from antiquity until about 1800”

Point (B)

We corrected Lines 749 & 750 of clean version

“(b) The Alterra-ILRI Publisher for their kindness in granting permission to re-publish Figures 6(f, g, h, i, ia), 23 and 24.”instead of“(b) The Alterra-ILRI Publisher for their kindness in granting permission to re-publish Figures 6(f, g, h, i), 23 and 24.” All changes are tracked in the revised version and a “clean” copy is also provided. On behalf of the co-authors

Yours sincerely,

Dr. Stavros Yannopoulos

Emeritus Professor
